# COPYRIGHT-PROTECTED LANGUAGE GENERATION VIA ADAPTIVE MODEL FUSION

**Javier Abad**
ETH Zurich

**Konstantin Donhauser**
ETH Zurich

**Francesco Pinto**[*]
University of Chicago

**Fanny Yang**[*]
ETH Zurich

## ABSTRACT

The risk of language models reproducing copyrighted material from their training data has led to the development of various protective measures. Among these, inference-time strategies that impose constraints via post-processing have shown promise in addressing the complexities of copyright regulation. However, they often incur prohibitive computational costs or suffer from performance trade-offs. To overcome these limitations, we introduce Copyright-Protecting Model Fusion (CP-Fuse), a novel approach that combines models trained on disjoint sets of copyrighted material during inference. In particular, CP-Fuse adaptively aggregates the model outputs to minimize the reproduction of copyrighted content, adhering to a crucial *balancing property* to prevent the regurgitation of memorized data. Through extensive experiments, we show that CP-Fuse significantly reduces the reproduction of protected material without compromising the quality of text and code generation. Moreover, its post-hoc nature allows seamless integration with other protective measures, further enhancing copyright safeguards. Lastly, we show that CP-Fuse is robust against common techniques for extracting training data[1].

## 1 INTRODUCTION

Large Language Models (LLMs), such as GPT-4 (Achiam et al., 2023) and Gemini (Team et al., 2023), have achieved undeniable success. However, they also present a significant challenge: the risk of reproducing copyrighted material from their training data (Sag, 2019; Yu et al., 2023; Karamolegkou et al., 2023). With the widespread adoption of these models for generating new text and code, the risk of large-scale copyright violations has become a pressing issue, as highlighted by several recent multi-million dollar lawsuits (Henderson et al., 2023). Consequently, preventing copyright infringement in language models has become a critical focus for researchers and practitioners alike.

The *fair use* doctrine (17 U.S.C. 107) offers a legal framework to mitigate liability for copyright infringement, spurring interest in techniques that allow models to use protected works without harming their market value (Henderson et al., 2023). Since verbatim reproduction often stems from memorization during training (Carlini et al., 2021; 2023; Somepalli et al., 2022; Pinto et al., 2024), researchers have explored training-time strategies that prevent memorization of individual data samples, such as Differentially Private (DP) training (Dwork et al., 2014; Abadi et al., 2016). However, scaling DP training to large models is computationally prohibitive and can degrade performance (Anil et al., 2022), motivating the exploration of alternative heuristic methods (Hans et al., 2024).

Beyond training-time approaches, pre-processing and inference-time methods have also been considered. Pre-processing strategies curate the training data by excluding or deduplicating copyrighted samples (Kandpal et al., 2022; Kocetkov et al., 2022; Min et al., 2023). However, these techniques are only partially effective (Lee et al., 2023; Ippolito et al., 2023) and often degrade model performance by removing high-quality copyrighted samples (Meeus et al., 2023). In contrast, inference-time methods intervene during decoding to prevent the reproduction of copyrighted content (Wei et al., 2024). Notably, Vyas et al. (2023) propose a general framework for constructing copyright-protected models by combining generative models trained on different data sources. While promising, their framework lacks a computationally feasible implementation suitable for real-world language models.

---

[*]Equal supervision

[1]See our GitHub repository: https://github.com/jaabmar/cp_fuse.

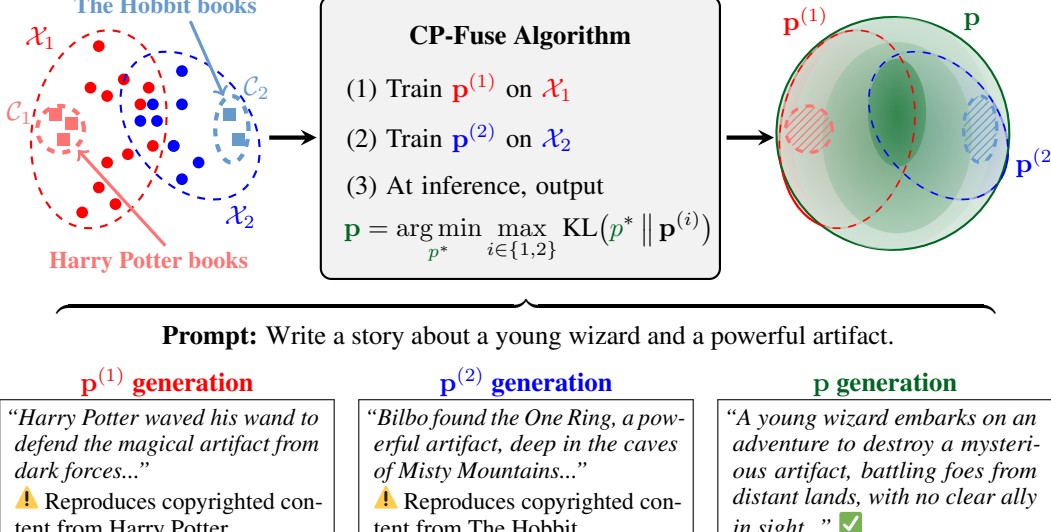

Figure 1: (**Top**) Illustration of the copyright-protecting fusion strategy. The left panel shows the training datasets $\mathcal{X}_1$ (in red) and $\mathcal{X}_2$ (in blue), each containing disjoint copyright sets $\mathcal{C}_1 \subset \mathcal{X}_1$ (in light red) and $\mathcal{C}_2 \subset \mathcal{X}_2$ (in light blue). The middle panel depicts our copyright-protecting fusion algorithm. The right panel displays the learned distributions of the potentially infringing models $\mathbf{p}^{(1)}$ (in red) and $\mathbf{p}^{(2)}$ (in blue), along with the resulting safe model $\mathbf{p}$ (in green). Lighter regions indicate areas of lower probability; although the safe model retains "access" to the copyrighted content, the probability of regurgitating it is very low. (**Bottom**) Generations from $\mathbf{p}^{(1)}$, $\mathbf{p}^{(2)}$, and $\mathbf{p}$ given the same prompt; the first two generations reproduce copyrighted material, $\mathbf{p}$ generates an original story.

In this paper, we propose **Copyright-Protecting Model Fusion (CP-Fuse)**, a simple yet effective method that combines outputs from multiple language models, each trained on disjoint sets of copyrighted material, to protect against infringement. Our approach builds on a rich body of work in model fusion for language models (Liu et al., 2021; Wang et al., 2023; Wan et al., 2024). Specifically, CP-Fuse adaptively aggregates logits to minimize the likelihood of reproducing copyrighted content (see Figure 1 for an illustration of our approach on a toy example). In Section 3.2, we first demonstrate that our strategy satisfies a *balancing property* (Lemma 3.3), which intuitively helps mitigate the regurgitation of memorized training samples. We then empirically show CP-Fuse's effectiveness in reducing regurgitation across various metrics that measure exact and approximate memorization. Notably, it reduces the reproduction of copyrighted material by over $25\times$ and consistently outperforms other inference-time methods while preserving the utility of the generated text and code (Section 4.2). Additionally, we demonstrate that applying CP-Fuse in combination with standard training-time methods further improves copyright protection (Section 4.3). Finally, we present experiments where CP-Fuse exhibits robustness against strategies that extract training data via prompting (Section 4.4).

## 2    RELATED WORKS ON COPYRIGHT PROTECTION

Copyright protection measures can be implemented at various stages of the model supply chain (Lee et al., 2023). In this section, we provide an overview of existing measures.

**Data pre-processing stage**    Many open-source LLMs are trained on datasets containing copyright-protected material, such as the BookCorpus dataset (e.g., GPT-3 (Brown et al., 2020)) and the C4 corpus (e.g., LLaMa (Touvron et al., 2023)). Therefore, efforts have been made to curate datasets with exclusively licensed content (Kocetkov et al., 2022; Min et al., 2023; Ippolito & Yu, 2023). Moreover, removing duplicate copyrighted samples from the dataset has been shown to reduce their regurgitation (Lee et al., 2022; Kandpal et al., 2022; Carlini et al., 2023). However, these approaches can be resource-intensive and degrade model performance (Ippolito et al., 2023; Meeus et al., 2023).

**Pre-training and fine-tuning stage** Other approaches intervene during the training or fine-tuning of LLMs to prevent memorization (Somepalli et al., 2022; Carlini et al., 2023; Zhang et al., 2023; Pinto et al., 2024; Aerni et al., 2024), ensuring that the distribution of the trained model assigns negligible probability to verbatim reproduction of training data. By design, differentially private (DP) methods (Dwork et al., 2014; Abadi et al., 2016) limit the influence of individual training points on the model's output and have thus been proposed to mitigate memorization issues. However, scaling DP training to LLMs with billions of parameters is computationally challenging and is associated with significant utility degradation (Anil et al., 2022). Furthermore, DP guarantees may lose relevance if samples are duplicated in the training set, and it has been argued that DP's goals differ from those of copyright protection (Elkin-Koren et al., 2023). Alternatively, heuristic methods, such as the goldfish loss (Hans et al., 2024) or early-stopping (Mireshghallah et al., 2022; Zhang et al., 2023; Pinto et al., 2024), have been empirically successful in reducing the likelihood of regurgitating training data.

**Inference and post-training stage** An orthogonal line enforces copyright constraints via post-processing (Wei et al., 2024). Filtering strategies, such as MemFree (Ippolito & Yu, 2023), can prevent the verbatim reproduction of copyrighted material from a curated blocklist during inference. However, these methods are effective only for exact verbatim matches and may lead to hallucinations due to modifications in the decoding process (Liu et al., 2024b). Other works propose unlearning copyrighted content from trained models (Bourtoule et al., 2021; Chen & Yang, 2023; Eldan & Russinovich, 2023; Jang et al., 2023; Zhang et al., 2024a; Liu et al., 2024a). However, these approaches are typically computationally impractical and require access to model weights, which is often restrictive in real-world scenarios. Our method draws inspiration from the Near-Access Free (NAF) framework (Vyas et al., 2023) and, unlike purely heuristic approaches, offers a principled theoretical explanation of how it prevents regurgitation (Lemma 3.3). We validate its effectiveness and competitiveness against baselines in extensive real-world experiments with popular language models (Section 4).

## 3 Copyright-Protecting Model Fusion

We focus on language models $p$ that take a prompt $x$ as input and return a probability distribution over a sequence of tokens of variable length $T$ from a fixed alphabet $V$, with $y_T = $ EOS representing the end-of-sequence token. Using the convention that $y_{<0} = \emptyset$, we can factorize $p$ as:

$$p(y_{0:T}|x) = \prod_{t=0}^{T} p(y_t|y_{<t}, x).$$

We now introduce a key assumption and background underlying our work.

### 3.1 Preliminaries

At the core of our method is the assumption of the *separability of copyrighted material*, as discussed by Vyas et al. (2023) for various vision and language applications. This assumption is akin to those used in exact machine unlearning (Bourtoule et al., 2021; Yan et al., 2022; Dukler et al., 2023) and in works that split datasets into safe and unsafe parts (Golatkar et al., 2021; 2024; Li et al., 2024).

Consider a dataset $\mathcal{D}$ and a set of copyright-protected material $\mathcal{C}$ that could be compromised when training a model $p$ on $\mathcal{D}$. The assumption states that we can split $\mathcal{D}$ into two potentially overlapping subsets, $\mathcal{D}_1$ and $\mathcal{D}_2$, such that each subset contains data associated with two non-overlapping sets of copyright-protected materials, $\mathcal{C}_1$ and $\mathcal{C}_2$, where $\mathcal{C}_1 \cap \mathcal{C}_2 = \emptyset$. This assumption holds, for instance, when we construct $\mathcal{D}$ from sources that are sufficiently distinct. Hence, any language model trained on the subset $\mathcal{D}_1$ is protected from infringing the copyright of materials in $\mathcal{C} \setminus \mathcal{C}_1 \supseteq \mathcal{C}_2$.

Given the two subsets $\mathcal{D}_1$ and $\mathcal{D}_2$, we can then train two generative models $p^{(1)}, p^{(2)}$ on the respective subsets and "fuse" them to construct a model $p$ that achieves protection against all copyright-protected material $\mathcal{C}$. In this context, Vyas et al. (2023) propose the $k$-NAF (Near Access Free) framework as a quantitative guarantee for copyright protection.

**Definition 3.1.** Formally, a model $p(.|x)$ is $k$-NAF for some $k \in \mathbb{R}^+$ if, for any input prompt $x$ and some user-specified divergence function $\Delta$,

$$\forall x: \quad \max_{i \in \{1,2\}} \Delta(p(.|x) \,||\, p^{(i)}(.|x)) \leq k. \tag{1}$$

The key intuition behind why a model satisfying $k$-NAF is less likely to regurgitate copyrighted material is as follows: if the separability of copyrighted material holds, the likelihood of generating infringing text for any material $c \in \mathcal{C}$ decreases exponentially with the length of $c$ for at least one of the models. Therefore, for a model $p$ to satisfy the $k$-NAF guarantee, it must assign a minimal probability to events involving the reproduction of protected material.

## 3.2 ALGORITHM

We introduce **C**opyright-**P**rotecting Model **Fus**ion (CP-Fuse), a simple yet effective algorithm for copyright protection via model fusion. Inspired by the $k$-NAF framework, we aim to minimize the maximum KL-divergence in Equation (1). However, achieving this directly is computationally intractable; therefore, we propose an efficient approximate algorithm that iteratively optimizes $p(y_t|y_{<t}, x)$ given the history $p(y_{<t}|x)$. In Lemma 3.2, we show that leveraging the KL-divergence allows us to express the update rule in a model fusion form. Formally, we iteratively define

$$
\begin{aligned}
p(y_t \,|\, y_{<t}, x) &= \arg\min_{p^*} \max_i \mathbb{E}_{y_t \sim p^*} \log\left( \frac{p^*(y_t) p(y_{<t} \,|\, x)}{p^{(i)}(y_{\leq t} \,|\, x)} \right) \\
&= \arg\min_{p^*, t} t \quad \text{s.t.}
\end{aligned}
\tag{2}
$$

$$
\forall i : \; \mathrm{KL}(p^* || p^{(i)}(.|y_{<t}, x)) + \log\left( \frac{p(y_{<t} \,|\, x)}{p^{(i)}(y_{<t} \,|\, x)} \right) \leq t,
$$

which results in a convex optimization problem. Although solving this problem naively is still computationally intensive, we overcome this limitation using the following lemma:

**Lemma 3.2.** *The optimal solution $p(y_t \,|\, y_{<t}, x)$ of the optimization problem in Equation* (2) *satisfies*[2]

$$
\log p^*(y_t) = \alpha_t \log p^{(1)}(y_t|y_{<t}, x) + \beta_t \log p^{(2)}(y_t|y_{<t}, x) + \gamma_t
\tag{3}
$$

*for some $\alpha_t, \beta_t \geq 0, \gamma_t \in \mathbb{R}$.*

Equation (2) can therefore be solved efficiently by performing a grid search over the parameters $\alpha_t$ and $\beta_t$, with $\gamma_t$ chosen as a function of $\alpha_t$ and $\beta_t$ to ensure the total probability mass sums to 1.

**Related works on model fusion** Our method is reminiscent of an extensive body of work focused on knowledge fusion in language models, both at inference time (Liu et al., 2021; Jiang et al., 2023; Gururangan et al., 2023; Mavromatis et al., 2024) and through weight merging after training (Wortsman et al., 2022; Jin et al., 2023; Hsu et al., 2024; Yadav et al., 2024). In particular, the minimizer derived in Lemma 3.2 relates to the former approaches, which generally define a model $p$ at inference time by combining multiple models $p^{(1)}, \ldots, p^{(K)}$ via a weighted sum of their logits:

$$
\log p(y_t|y_{<t}, x) := \sum_{i=1}^K \alpha_t^{(i)}(y_{<t}, x) \log p^{(i)}(.|y_{<t}, x) + c,
\tag{4}
$$

where $c$ is a normalizing constant, and $\alpha_t^{(i)}$ can depend on both the prompt $x$ and the history $y_{<t}$. However, previous model fusion approaches are not designed to mitigate copyright infringement.

Vyas et al. (2023) propose CP-$\Delta$ as a general framework for combining two generative models aimed at achieving copyright protection. Yet, their approach becomes computationally intractable when applied directly to the probability distribution $p(.|x)$ over the entire sequence $y_T$. To address this, the authors suggest applying CP-$\Delta$ token-wise instead, resulting in the model from Equation (3) with $\alpha_t = \beta_t = 1/2$. However, they do not benchmark this method against other copyright-protection techniques. In the following section, we discuss why this approach is ineffective for copyright protection and how our method overcomes this issue by incorporating the sequence history $y_{<t}$.

## 3.3 EFFICACY OF METHODOLOGY

In this section, we provide an intuitive explanation of how our method prevents the reproduction of memorized training samples. Recall that, by Lemma 3.2, CP-Fuse adaptively selects $\alpha_t$ and $\beta_t$ based on the sequence history $y_{<t}$. Specifically, the algorithm assigns less weight to the model that has been more dominant in generating $y_{<t}$. More formally, the following balancing property holds:

---

[2]We set $\log(0) = -\infty$

**Lemma 3.3.** *(Balancing property) Given a prompt $x$, let $y_{<t}$ be any non-ending sequence and assume that $p^{(i)}(.|y_{<t}, x)$ has full support for $i \in \{1, 2\}$ and $p^{(1)}(y_{<t}|x) > p^{(2)}(y_{<t}|x)$. Then, either*

1.  $$\mathbb{E}_{y_t \sim p(.|y_{<t}, x)} \log p^{(1)}(y_{\leq t}|x) = \mathbb{E}_{y_t \sim p(.|y_{<t}, x)} \log p^{(2)}(y_{\leq t}|x)$$

2.  $$p(y_t|y_{<t}, x) = p^{(2)}(y_t|y_{<t}, x)$$

Intuitively, this balancing property ensures that neither model dominates the text generation. As an example, suppose the generation of a subsequence $y_{<t}$ is strongly dominated by $p^{(1)}$, such that $p^{(1)}(y_{<t}|x) \gg p^{(2)}(y_{<t}|x)$. If the first case in Lemma 3.3 holds, the output distribution of the copyright-protected model, $p(y_t|y_{<t}, x)$, will be such that, in expectation, the sequence with the new token has the same log-probability for both models, i.e., loosely speaking $p^{(1)}(y_{\leq t}|x) \approx p^{(2)}(y_{\leq t}|x)$. Conversely, if the second case $p = p^{(2)}$ holds, then the generation of $y_t$ is independent of $p^{(1)}(y_{<t}|x)$. In other words, the next token generated by $p$ will likely match the most probable token under the dominant model, $p^{(1)}(y_{<t}|x)$, only if both $p^{(1)}$ and $p^{(2)}$ are close conditioned on $y_{<t}$ and $x$, that is, when the generated se-

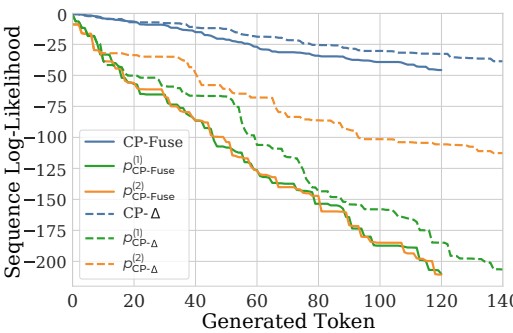

Figure 2: Log-likelihood of sequences produced by CP-Fuse and CP-$\Delta$, and their base models $p^{(1)}$ and $p^{(2)}$, at each generated token. We show a random generation from StarCoder fine-tuned on the `Python instructions`; details in Section 4.

quence is not protected assuming separability of copyrighted material (Section 3.1). We provide additional experimental evidence for this property in Appendix A.7.

We now illustrate how adaptively choosing $\alpha_t$ and $\beta_t$ is crucial for achieving effective copyright protection. We present in Figure 2 the cumulative log-likelihood at each generated token for sequences produced by CP-Fuse and token-wise CP-$\Delta$ (which does not consider the history), alongside their respective base models $p^{(1)}$ and $p^{(2)}$. The balancing property of CP-Fuse ensures that the log-likelihoods under $p^{(1)}$ and $p^{(2)}$ are approximately equal for the generated sequence at each token, thereby preventing the reproduction of copyrighted material, as no protected content is memorized by both base models. In contrast, CP-$\Delta$ shows a clear preference for $p^{(2)}$, suggesting that $p^{(2)}$ may have memorized protected training samples, making CP-$\Delta$ vulnerable to reproducing them. In the next section, we further validate this observation through extensive real-world experiments.

## 4 EXPERIMENTS

We conduct our experiments using language models that are commonly employed in practical applications. Each dataset (details provided below) is partitioned into two non-overlapping subsets of 3,000 samples each, and a separate model is fine-tuned on each subset. In this setup, every training example is treated as potentially copyright-protected. To evaluate the copyright protection capabilities of CP-Fuse, we simulate an extreme scenario by overfitting the models to the subsets by fine-tuning them for 50 epochs (see Appendix D). As a result, the base models strongly memorize the data, representing a challenging setting where they are prone to reproducing any training sample.

**Datasets and Models**  We perform fine-tuning using four datasets, each associated with a specific task. **(I) Abstract Generation:** We fine-tune the LLaMa2 7B model (Touvron et al., 2023) on a dataset of abstracts from math papers (`MathAbstracts`) (Zhang et al., 2024b), using each paper's title as the prompt. **(II) Story-telling:** We also fine-tune LLaMa2 on the story-generation dataset `WritingPrompts` (Fan et al., 2018), using the topic of the story as the prompt. **(III) Code Generation:** We fine-tune the StarCoder 7B model (Li et al., 2023) on an instructional dataset for Python (`Python instructions`), where the prompts are natural language descriptions of tasks and the responses are Python solutions. **(IV) Code Generation with Unit Tests:** We fine-tune again a StarCoder model on the `APPS` dataset (Hendrycks et al., 2021), which also consists of natural language problems and Python solutions, but additionally incorporates unit tests to assess code

generation quality. For this task, the models are further evaluated on the `MBPP` (Austin et al., 2021) and `HumanEval` (Chen et al., 2021) datasets, which also include unit tests. Both code and text-based tasks represent settings where copyright infringement is a concern (Yu et al., 2023; Henderson et al., 2023). We provide additional information on the datasets in Appendix D.4.

**Copyright-infringement metrics** We use a comprehensive set of metrics to evaluate potential copyright infringement. The models are prompted with the prefix of samples from their fine-tuning splits. We then compare the generated outputs against the original samples. For each metric, we report the average value above the 95th percentile. This focus on high percentiles is motivated by the legal concern that models might reproduce only a few long text extracts in a real-world scenario.

To measure *exact memorization*, we use the average Exact Matching (EM) length and the Infringement Count (IC) for substrings exceeding 160 characters (in Appendix A.1). While the former is widely recognized in the literature as a clear indicator of copyright infringement in both text and code (Lee et al., 2022; Karamolegkou et al., 2023; Carlini et al., 2023; Yu et al., 2023), the latter count with a threshold at 160 is consistent with regulatory guidelines (Mueller et al., 2024).

For *approximate memorization*, we report the BLEU score and normalized Levenshtein distance in the main text, with additional results for ROUGE-L score, METEOR score, Jaccard similarity, cosine similarity, and semantic similarity in Appendix A.1. These metrics are well-established in the literature, see e.g., (Ippolito et al., 2023; Huang et al., 2023; Chen et al., 2024). For the `Python instructions` dataset, we also report specialized metrics obtained from two state-of-the-art plagiarism detection tools: JPlag (Prechelt et al., 2002) and Dolos (Maertens et al., 2022). We refer to Appendix D.6.1 for full details on the implementation of the copyright-infringement metrics.

**Utility metrics** For text-based tasks, we evaluate *fluency* using the Prometheus-v2 model (Kim et al., 2024), which serves as a judge for the stories generated by models fine-tuned on `WritingPrompts`. Model-based fluency metrics have been shown to closely align with human evaluations (Liu et al., 2023; Sottana et al., 2023), particularly for the `WritingPrompts` dataset (Chiang & Lee, 2023). Prometheus-v2 has also demonstrated consistent alignment with human annotators and GPT-4 (Kim et al., 2024). We use the five-point rubric detailed in Figure 21. For code-based tasks, we use the `APPS`, `MBPP`, and `HumanEval` datasets, which include unit tests that allow for the computation of the *pass@1 score* (Chen et al., 2021). We report the utility metrics in a separate test set comprising 500 prompts for each dataset. Refer to Appendix D.6.2 for full details on the implementation.

**Comparison with inference-time baselines** We compare CP-Fuse against three protection measures that intervene at the inference stage. We report results using token-wise CP-$\Delta$ (Vyas et al., 2023), with KL divergence as $\Delta$; a system-mode self-reminder method (Xie et al., 2023), where the model is explicitly instructed to avoid regurgitating memorized data; and MemFree decoding (Ippolito et al., 2023), which prevents 10-gram copying from the training data by using a Bloom filter. Implementation details for these baselines are provided in Appendix D.5. For CP-Fuse, we construct the grid by discretizing the interval $[0, 2)$ into 10 steps, and $[2, 10]$ into 9 steps (see ablation studies in Appendix A.6). Greedy decoding is used for all experiments presented in the main text, with results from temperature sampling—leading to equivalent conclusions—provided in Appendix A.4. Additionally, we conduct experiments combining three models instead of two in Appendix A.8.

**Combining CP-Fuse with training-time strategies** In Section 4.3, we apply CP-Fuse on top of models trained with the goldfish loss (Hans et al., 2024), setting the dropout frequency to $k = 16$. In Appendix A.5, we use CP-Fuse to wrap early-stopped models, a simple way of preventing memorization (Mireshghallah et al., 2022). Here, we stop fine-tuning after 3 epochs.

### 4.1 PREVENTING COPYRIGHT INFRINGEMENT WITH CP-FUSE

Table 1 presents the copyright-infringement metrics for the overfitted models, the baselines, and CP-Fuse across fine-tuning splits. CP-Fuse substantially reduces verbatim regurgitation in both code and text tasks. Specifically, CP-Fuse decreases exact matches by more than a factor of 25 compared to the overfitted models and consistently outperforms the baselines. In particular, the system-mode self-reminder (SystemPrompt) fails to prevent the reproduction of memorized samples, possibly due to the override of safety fine-tuning during our fine-tuning. MemFree noticeably reduces the length

Table 1: Copyright-infringement metrics **averaged at the 95th percentile** for the `Python instructions`, `MathAbstracts`, and `WritingPrompts` datasets across fine-tuning splits. We present results for the overfitted models, Self-reminder prompting (SystemPrompt), MemFree, CP-$\Delta$, and CP-Fuse. Metrics include Exact Matching (EM), BLEU score (BLE), Levenshtein Distance (LEV), and code plagiarism score JPlag (JP). Arrows indicate the preferred direction: $\uparrow$ (higher is better) and $\downarrow$ (lower is better). We highlight in **bold** the best method for each split and metric.

| Model | Split | EM$\downarrow$ | JP$\downarrow$ | EM$\downarrow$ | BLE$\downarrow$ | LEV$\uparrow$ | EM$\downarrow$ | BLE$\downarrow$ | LEV$\uparrow$ |
|---|---|---|---|---|---|---|---|---|---|
| | | `Python inst.` | | `MathAbstracts` | | | `WritingPrompts` | | |
| Overfit | Split 1 | 1469.80 | 1.00 | 1397.68 | 1.00 | 0.00 | 1316.24 | 1.00 | 0.00 |
| Split 1 | Split 2 | 44.60 | 0.01 | 31.00 | 0.01 | 0.70 | 22.59 | 0.03 | 0.71 |
| Overfit | Split 1 | 42.64 | 0.01 | 42.12 | 0.08 | 0.68 | 26.11 | 0.02 | 0.70 |
| Split 2 | Split 2 | 1393.88 | 0.99 | 1570.88 | 1.00 | 0.00 | 1141.88 | 1.00 | 0.00 |
| System | Split 1 | 1360.68 | 1.00 | 1022.72 | 0.99 | 0.00 | 1118.36 | 1.00 | 0.00 |
| Prompt | Split 2 | 1373.16 | 0.99 | 1005.20 | 1.00 | 0.00 | 1092.20 | 1.00 | 0.00 |
| MemFree | Split 1 | 165.48 | 0.99 | 111.12 | 0.23 | 0.53 | 63.84 | 0.20 | 0.55 |
| | Split 2 | 157.36 | 0.96 | 99.40 | 0.22 | 0.53 | 59.04 | 0.19 | 0.55 |
| CP-$\Delta$ | Split 1 | 273.20 | 1.00 | 341.60 | 0.58 | 0.30 | 37.40 | 0.05 | **0.70** |
| | Split 2 | 284.80 | 0.99 | 162.80 | 0.30 | 0.51 | 31.29 | 0.04 | **0.70** |
| CP-Fuse | Split 1 | **69.58** | **0.03** | **55.54** | **0.14** | **0.62** | **27.55** | **0.02** | **0.70** |
| (Ours) | Split 2 | **68.04** | **0.03** | **48.74** | **0.14** | **0.63** | **25.50** | **0.03** | **0.70** |

of memorized text segments; however, it still reproduces segments twice as long as those produced by CP-Fuse. A quick inspection of MemFree outputs also reveals that it often avoids exact copying simply by inserting spaces or spelling mistakes (refer to Appendix A.10 for examples). Finally, CP-$\Delta$ still generates long text segments that exactly match the training data, often 4 to 7 times longer than those produced by CP-Fuse. These segments frequently exceed the legal 160-character threshold, while such infringements almost entirely disappear with our method (refer to Appendix A.1).

The approximate memorization metrics further support these observations. We observe a consistent improvement with CP-Fuse compared to the overfitted models, also offering better protection than the baselines. Notably, CP-Fuse achieves near "non-plagiarism" on the code task, as the JPlag tool detects almost no copying in the generated code, while it clearly indicates infringement for all baselines.

These findings demonstrate the effectiveness of our method in preventing both verbatim and quasi-verbatim reproduction of training material. Additional metrics supporting these conclusions are provided in Appendix A.1. Furthermore, experiments using temperature sampling decoding, detailed in Appendix A.4, yield equivalent conclusions. We also explore combining three models instead of two in Appendix A.8, which provides further improvement in the memorization metrics.

**Comparison between CP-Fuse and CP-$\Delta$** We now provide a more detailed comparison between our method and CP-$\Delta$. Figure 3 shows histograms illustrating the distribution of exact matches generated by both methods. We observe a significantly heavier-tailed distribution for CP-$\Delta$, which consistently reproduces longer verbatim text segments than CP-Fuse and is therefore more likely to infringe on copyright. For instance, the longest exact match for CP-Fuse in the abstracts task is 73 characters, whereas the 95th percentile for CP-$\Delta$ is 342 characters, with the longest match exceeding 500 characters. These results highlight the importance of adaptively setting the weights based on sequence history when combining the models to ensure effective copyright protection. We provide examples of extracts produced by both methods for different datasets in Appendix A.9.

## 4.2 UTILITY EVALUATION FOR CODE GENERATION AND STORY-TELLING

The success of our method in preventing the exact and approximate reproduction of training samples raises the question of whether it still generates useful text and code for solving the task at hand. In Table 2, we present the results for the considered utility metrics, pass@1 for code-based tasks and fluency for story-telling writing, on the test splits not seen during fine-tuning. We observe that

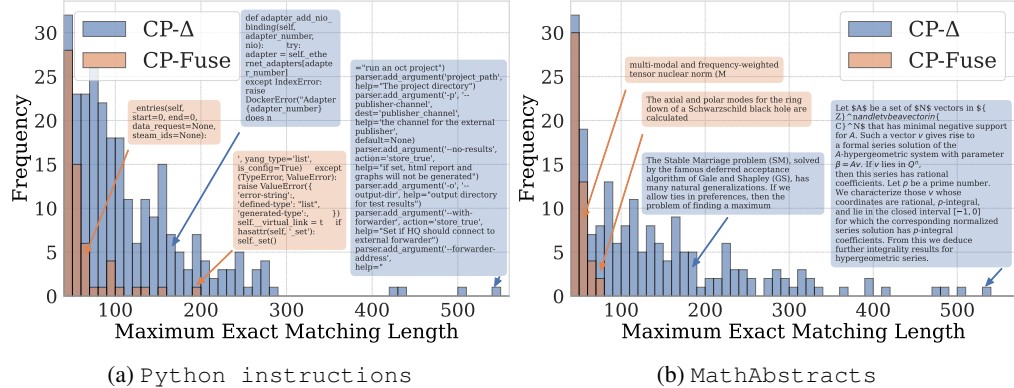

(a) `Python instructions`  (b) `MathAbstracts`

Figure 3: Histogram of exactly matched substring lengths (above 40 characters) generated by CP-$\Delta$ and CP-Fuse for (a) the `Python instructions` and (b) the `MathAbstracts` datasets. We show the longest substring and one randomly sampled match above 40 characters.

CP-Fuse produces code that is as accurate as that generated by the overfitted models, passing a similar proportion of unit tests. Additionally, the stories generated based on the `WritingPrompts` dataset achieve the same level of fluency as those produced by copyright-infringing models. All baselines perform similarly, with the notable exception of MemFree, which consistently underperforms.

We further validate the high quality of text and code produced by our method with examples of its outputs in Appendix A.9. For the code tasks, CP-Fuse generates code that is significantly different from the original, effectively solving the tasks while often incorporating exception handling and additional features. For the text tasks, it produces well-written stories and coherent abstracts that also differ notably from the reference material.

**Comparison between CP-Fuse and MemFree**  Filtering approaches like MemFree are susceptible to hallucinations (Ippolito et al., 2023; Liu et al., 2024b). This issue is particularly problematic in code generation, where even minor changes, such as modifying variable names or introducing typos, can result in incorrect code. As a result, MemFree significantly underperforms on the `APPS` dataset due to the lengthy Python solutions, which increase the likelihood of hallucinations and, consequently, errors. We provide two examples in Figure 4. The `MBPP` and `HumanEval` datasets, with their shorter problems, are less affected by this issue. A similar problem arises in the `WritingPrompts` dataset, where alterations in the decoding process can lead to grammatical and spelling errors or cause the model to deviate from the intended topic (see Appendix A.10 for examples). In contrast, CP-Fuse generates tokens that are consistent with at least one of the two combined models.

Table 2: Utility metrics across datasets. Pass@1 is reported for `APPS`, `MBPP`, and `HumanEval` (`HE`); Fluency is reported for `WritingPrompts` (`WP`). Arrows indicate the preferred direction.

|  | **Pass@1** ↑ | | | **Fluency** ↑ |
|---|---|---|---|---|
|  | `APPS` | `MBPP` | `HE` | `WP` |
| Overfit Split 1 | 0.43 | 0.44 | 0.29 | 2.17 |
| Overfit Split 2 | 0.42 | 0.44 | 0.28 | 2.16 |
| SystemPrompt | 0.41 | 0.43 | 0.29 | 2.00 |
| MemFree | 0.32 | 0.41 | 0.24 | 1.70 |
| CP-$\Delta$ | 0.45 | 0.46 | 0.29 | 2.15 |
| CP-Fuse (Ours) | 0.47 | 0.43 | 0.28 | 2.17 |

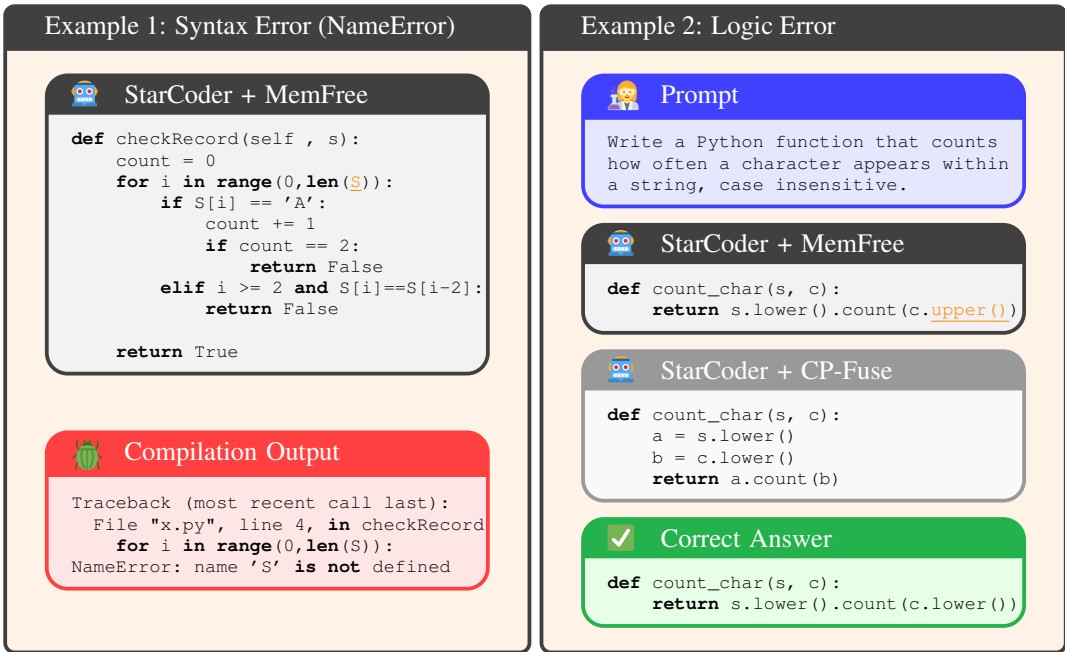

Figure 4: Examples of typical errors in code generated by MemFree for the `APPS` dataset. The highlighted characters indicate the code affected by filtering, leading to errors. On the left, MemFree changes a variable name, resulting in a syntax error. On the right, MemFree alters the logic of the code, producing incorrect output, whereas CP-Fuse successfully solves the problem.

## 4.3 CP-FUSE AS A WRAPPER OF OTHER PROTECTION METHODS

In this section, we demonstrate how CP-Fuse can be combined with other training-time methods to improve protection. Specifically, we apply CP-Fuse on top of models fine-tuned on the `WritingPrompts` dataset using the goldfish loss (Hans et al., 2024), a recently proposed technique designed to mitigate memorization during training. As presented in Figure 5a, integrating CP-Fuse results in further improvements in both the exact match and BLEU score. We also conduct similar experiments in Appendix A.5, where CP-Fuse is applied on top of early-stopped models—another strategy aimed at reducing memorization during training.

## 4.4 ROBUSTNESS OF CP-FUSE UNDER PREFIX PROMPTING EXTRACTIONS

Previous sections have shown that our method effectively reduces regurgitation without sacrificing utility. In this final section, we evaluate the robustness of CP-Fuse against *prefix prompting* extractions using the `WritingPrompts` dataset. We assume a *threat model* where an adversary has black-box access to CP-Fuse, which wraps two potentially copyright-infringing models. The attacker has access to the prompts used during fine-tuning, as well as a prefix of the original story, and their *goal* is to have CP-Fuse regurgitate a story memorized by one of the base models. We study the impact of increasing the prefix length on the exact match (EM) and BLEU score of the generated outputs. The results in Figure 5b show that CP-Fuse remains robust as the prefix length increases. Specifically, exact matching remains relatively stable, while there is a slight increase in the BLEU score; however, it is still significantly smaller than that observed in the overfitted models and baselines (Table 1). In Appendix A.11, we visualize the stories generated by CP-Fuse and show how they evolve given short and long prefix lengths. CP-Fuse consistently produces outputs that differ from the reference, thanks to its balancing mechanism that prevents any single model from dominating the generation process.

| Model | Split | EM$\downarrow$ | BLE$\downarrow$ |
|---|---|---|---|
| GL Split 1 | Split 1 | 84.68 | 0.11 |
| | Split 2 | 21.79 | 0.03 |
| GL Split 2 | Split 1 | 19.11 | 0.02 |
| | Split 2 | 120.28 | 0.16 |
| CP-Fuse | Split 1 | 20.68 | 0.03 |
| | Split 2 | 25.50 | 0.03 |

(a) Wrapping experiments with GL loss

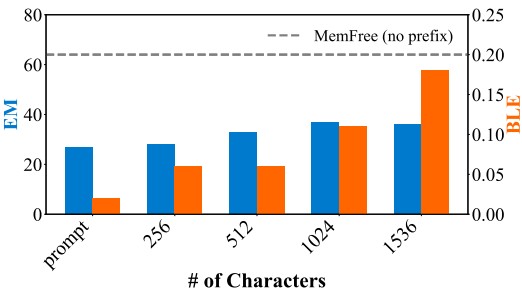

(b) Prefix prompting experiments

Figure 5: Metrics for copyright infringement, exact matching (EM), and BLEU score (BLE) in the `WritingPrompts` dataset for (a) models trained with goldfish loss (GL) and CP-Fuse as a wrapper, and (b) the effect of the prefix length on CP-Fuse applied to overfitted models in the split 1.

## 5 CONCLUSIONS

In this paper, we introduced CP-Fuse, a simple yet effective algorithm for copyright protection based on model fusion. We first demonstrated that CP-Fuse satisfies key properties for preventing the reproduction of memorized training samples. Additionally, we provided extensive evidence of its effectiveness in challenging scenarios involving overfitted models, where CP-Fuse significantly reduced the regurgitation of protected materials, outperforming other inference-time baselines without compromising the quality of the generated code and text. The versatility of CP-Fuse was demonstrated through its seamless integration with other training-time techniques that mitigate memorization. Finally, we showcased its robustness against standard prefix prompting extraction strategies.

It is important to note that CP-Fuse operates under the assumption of copyright separability. A potential avenue for future research is theoretically investigating CP-Fuse's performance when this assumption only partially holds. Moreover, it would be valuable to apply our algorithm in real-world scenarios involving larger models and genuine copyright-protected content, such as books or songs.

## ACKNOWLEDGEMENTS

JA and KD acknowledge support from the ETH AI Center. KD further acknowledges support from the ETH Foundations of Data Science. This work was done in part while FY was visiting the Simons Institute for the Theory of Computing. We extend our gratitude to Javier Rando for insightful discussions and valuable feedback on the manuscript.

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

TABLE OF CONTENTS

APPENDICES

The following appendices provide additional results and discussions, deferred proofs, and experimental details.

## A ADDITIONAL EXPERIMENTS

### A.1 ADDITIONAL METRICS FOR COPYRIGHT INFRINGEMENT

First, we report in Table 1 the full results with JPlag (Prechelt et al., 2002) and Dolos (Maertens et al., 2022)—the specialized software plagiarism metrics—for the `Python instructions` dataset. The additional results with Dolos are consistent with those of JPlag, which were already reported in the main text; that is, CP-Fuse shows the lowest score and is hence the least likely to infringe on copyright in code generation.

We also present the complete results for the copyright-infringement metrics for StarCoder in the `Python instructions` dataset (Table 4), and for the LLaMa2 model in the `MathAbstracts` (Table 5) and `WritingPrompts` (Table 6) datasets, along with additional metrics. Specifically, we include Jaccard and cosine similarities, and the METEOR score to measure approximate memorization, as well as semantic similarity for a higher-level measure that does not necessarily indicate copyright infringement. We report results for the overfitted models, the baselines, and CP-Fuse. We include additional results on a test set comprising 500 prompts.

Results using Jaccard and cosine similarities and the METEOR score confirm the observations from the main text, closely aligning with previous metrics: CP-Fuse is consistently the best or (less often) second-best method. The semantic similarity for CP-Fuse and baselines remains consistently high, comparable to that of the overfitted models, suggesting that no semantic information is lost when applying these methods.

Finally, for completeness, Table 7 shows the exact matching above the 95th percentile for StarCoder models trained on the `APPS` dataset. CP-Fuse continues to be effective in reducing regurgitation in this setting.

Table 3: Dolos and JPlag plagiarism metrics for the `Python instructions` dataset. ↓ Means lower is better, we highlight in **bold** the best method for each split and metric.

| Model | Split | JPlag↓ | Dolos↓ |
|---|---|---|---|
| Overfit Split 1 | Split 1 | 1.00 | 1.00 |
| | Split 2 | 0.01 | 0.35 |
| | Test | 0.02 | 0.39 |
| Overfit Split 2 | Split 1 | 0.01 | 0.40 |
| | Split 2 | 0.99 | 1.00 |
| | Test | 0.01 | 0.32 |
| System Prompt | Split 1 | 1.00 | 1.00 |
| | Split 2 | 0.99 | 1.00 |
| | Test | **0.01** | 0.34 |
| MemFree | Split 1 | 0.99 | 0.98 |
| | Split 2 | 0.96 | 0.99 |
| | Test | **0.01** | **0.27** |
| CP-$\Delta$ | Split 1 | 1.00 | 1.00 |
| | Split 2 | 0.99 | 1.00 |
| | Test | **0.01** | 0.37 |
| CP-Fuse (Ours) | Split 1 | **0.03** | **0.70** |
| | Split 2 | **0.03** | **0.71** |
| | Test | **0.01** | 0.37 |

Table 4: Copyright-infringement metrics averaged at the 95th percentile for StarCoder in the Python instruction dataset across different data splits. The table presents results for the overfitted models, SystemPrompt, MemFree, CP-$\Delta$, and CP-Fuse. Metrics include Exact Matching (EM), Normalized Levenshtein Distance (LEV), Jaccard Similarity (JAC), Cosine Similarity (COS), Semantic Similarity (SEM), ROUGE-L (ROU), BLEU Score (BLE), METEOR Score (MET), and Infringement Count ($IC_{50}$, $IC_{160}$). $\downarrow$ Means lower is better, $\uparrow$ means higher is better.

| | | EM $\downarrow$ | $IC_{50}$ $\downarrow$ | $IC_{160}$ $\downarrow$ | ROU $\downarrow$ | BLE $\downarrow$ | MET $\downarrow$ | JAC $\downarrow$ | COS $\downarrow$ | SEM $\downarrow$ | LEV $\uparrow$ |
|---|---|---|---|---|---|---|---|---|---|---|---|
| **Python instructions** | | | | | | | | | | | |
| Overfit Split 1 | Split 1 | 1469.80 | 1427.20 | 1310.80 | 1.00 | 1.00 | 1.00 | 1.00 | 1.00 | 1.00 | 0.00 |
| | Split 2 | 44.60 | 0.95 | 0.05 | 0.48 | 0.12 | 0.34 | 0.27 | 0.56 | 0.97 | 0.56 |
| | Test | 54.38 | 1.03 | 0.00 | 0.50 | 0.12 | 0.38 | 0.29 | 0.62 | 0.97 | 0.55 |
| Overfit Split 2 | Split 1 | 42.64 | 0.79 | 0.00 | 0.46 | 0.10 | 0.32 | 0.25 | 0.60 | 0.98 | 0.55 |
| | Split 2 | 1393.88 | 1380.96 | 1257.80 | 1.00 | 1.00 | 1.00 | 1.00 | 1.00 | 1.00 | 0.00 |
| | Test | 53.26 | 1.11 | 0.16 | 0.51 | 0.10 | 0.36 | 0.27 | 0.58 | 0.97 | 0.54 |
| System Prompt | Split 1 | 1360.68 | 1319.32 | 1201.68 | 1.00 | 0.99 | 1.00 | 1.00 | 1.00 | 1.00 | 0.01 |
| | Split 2 | 1373.16 | 1331.32 | 1214.16 | 1.00 | 1.00 | 1.00 | 1.00 | 1.00 | 1.00 | 0.00 |
| | Test | 72.44 | 1.73 | 0.50 | 0.50 | 0.14 | 0.37 | 0.29 | 0.60 | **0.97** | 0.52 |
| MemFree | Split 1 | 165.48 | 193.96 | 1.11 | 0.97 | 0.79 | 0.92 | 0.82 | 0.95 | 1.00 | 0.03 |
| | Split 2 | 157.36 | 190.81 | 0.65 | 0.98 | 0.77 | 0.92 | 0.82 | 0.94 | 1.00 | 0.03 |
| | Test | 52.33 | 0.72 | 0.02 | **0.46** | 0.10 | **0.33** | **0.27** | **0.58** | **0.97** | **0.59** |
| CP-$\Delta$ | Split 1 | 273.20 | 312.28 | 136.52 | 1.00 | 1.00 | 1.00 | 1.00 | 1.00 | 1.00 | 0.01 |
| | Split 2 | 284.80 | 337.28 | 152.64 | 1.00 | 1.00 | 1.00 | 1.00 | 1.00 | 1.00 | 0.02 |
| | Test | 57.50 | 1.20 | 0.03 | 0.58 | 0.12 | 0.40 | 0.31 | 0.61 | **0.97** | 0.50 |
| CP-Fuse (Ours) | Split 1 | **69.58** | **80.62** | **0.96** | **0.78** | **0.39** | **0.67** | **0.54** | **0.77** | **0.99** | **0.25** |
| | Split 2 | **68.04** | **2.426** | **0.23** | **0.77** | **0.37** | **0.66** | **0.52** | **0.74** | **0.99** | **0.30** |
| | Test | **40.00** | **0.40** | **0.00** | 0.52 | **0.09** | 0.34 | 0.29 | 0.59 | **0.97** | 0.55 |

Table 5: Copyright-infringement metrics (as in Table 4) for LLaMa2 models in the `MathAbstracts` dataset. $\downarrow$ Means lower is better, $\uparrow$ higher is better.

| `MathAbstracts` | | EM $\downarrow$ | IC$_{50}$ $\downarrow$ | IC$_{160}$ $\downarrow$ | ROU $\downarrow$ | BLE $\downarrow$ | MET $\downarrow$ | JAC $\downarrow$ | COS $\downarrow$ | SEM $\downarrow$ | LEV $\uparrow$ |
|---|---|---|---|---|---|---|---|---|---|---|---|
| Overfit Split 1 | Split 1 | 1397.68 | 1595.12 | 1482.84 | 1.00 | 1.00 | 1.00 | 1.00 | 1.00 | 1.00 | 0.00 |
| | Split 2 | 31.00 | 0.02 | 0.00 | 0.25 | 0.01 | 0.24 | 0.17 | 0.76 | 0.98 | 0.70 |
| | Test | 28.76 | 0.00 | 0.00 | 0.22 | 0.04 | 0.23 | 0.16 | 0.75 | 0.98 | 0.71 |
| Overfit Split 2 | Split 1 | 42.12 | 0.17 | 0.00 | 0.27 | 0.08 | 0.27 | 0.19 | 0.77 | 0.98 | 0.68 |
| | Split 2 | 1570.88 | 1798.72 | 1688.72 | 1.00 | 1.00 | 1.00 | 1.00 | 1.00 | 1.00 | 0.00 |
| | Test | 37.48 | 0.07 | 0.00 | 0.26 | 0.07 | 0.26 | 0.19 | 0.76 | 0.98 | 0.69 |
| System Prompt | Split 1 | 1022.72 | 1155.36 | 1040.24 | 1.00 | 0.99 | 1.00 | 1.00 | 1.00 | 1.00 | 0.00 |
| | Split 2 | 1005.20 | 1143.56 | 1024.76 | 1.00 | 1.00 | 1.00 | 1.00 | 1.00 | 1.00 | 0.00 |
| | Test | 36.44 | 0.07 | 0.00 | **0.25** | **0.06** | 0.27 | **0.19** | **0.75** | **0.98** | 0.69 |
| MemFree | Split 1 | 111.12 | 25.04 | **0.00** | 0.44 | 0.23 | 0.41 | 0.31 | **0.80** | 0.99 | 0.53 |
| | Split 2 | 99.40 | 29.31 | **0.00** | 0.45 | 0.22 | 0.39 | 0.31 | 0.82 | 0.99 | 0.53 |
| | Test | 39.32 | 0.09 | **0.00** | 0.26 | 0.07 | 0.28 | **0.19** | 0.76 | 0.99 | **0.70** |
| CP-$\Delta$ | Split 1 | 341.60 | 408.72 | 253.4 | 0.72 | 0.58 | 0.64 | 0.61 | 0.89 | 0.99 | 0.30 |
| | Split 2 | 162.80 | 162.60 | 1.66 | 0.50 | 0.30 | 0.40 | 0.35 | 0.86 | **0.98** | 0.51 |
| | Test | 39.91 | **0.01** | **0.00** | 0.29 | 0.07 | 0.26 | 0.21 | 0.77 | 0.98 | 0.67 |
| CP-Fuse (Ours) | Split 1 | **55.54** | **15.14** | **0.00** | **0.35** | **0.14** | **0.30** | **0.26** | **0.80** | **0.98** | **0.62** |
| | Split 2 | **48.74** | **0.378** | **0.00** | **0.34** | **0.14** | **0.31** | **0.24** | **0.81** | **0.98** | **0.63** |
| | Test | **35.59** | **0.01** | **0.00** | **0.28** | 0.07 | **0.25** | **0.19** | 0.77 | **0.98** | 0.68 |

Table 6: Copyright-infringement metrics (as in Table 4) for LLaMa2 models in the `WritingPrompts` dataset. ↓ Means lower is better, ↑ higher is better.

| | | EM ↓ | IC$_{50}$ ↓ | IC$_{160}$ ↓ | ROU ↓ | BLE ↓ | MET ↓ | JAC ↓ | COS ↓ | SEM ↓ | LEV ↑ |
|---|---|---|---|---|---|---|---|---|---|---|---|
| `WritingPrompts` | | | | | | | | | | | |
| Overfit Split 1 | Split 1 | 1316.24 | 1679.71 | 1569.71 | 1.00 | 1.00 | 1.00 | 1.00 | 1.00 | 1.00 | 0.00 |
| | Split 2 | 22.59 | 0.12 | 0.00 | 0.18 | 0.03 | 0.25 | 0.15 | 0.68 | 0.99 | 0.71 |
| | Test | 16.89 | 0.00 | 0.00 | 0.18 | 0.02 | 0.24 | 0.15 | 0.71 | 0.99 | 0.71 |
| Overfit Split 2 | Split 1 | 26.11 | 0.15 | 0.00 | 0.17 | 0.02 | 0.24 | 0.15 | 0.70 | 0.99 | 0.70 |
| | Split 2 | 1141.88 | 1385.84 | 1275.84 | 1.00 | 1.00 | 1.00 | 1.00 | 1.00 | 1.00 | 0.00 |
| | Test | 23.60 | 0.00 | 0.00 | 0.18 | 0.03 | 0.24 | 0.16 | 0.68 | 0.99 | 0.71 |
| System Prompt | Split 1 | 1118.36 | 1464.72 | 1250.52 | 1.00 | 1.00 | 1.00 | 1.00 | 1.00 | 1.00 | 0.00 |
| | Split 2 | 1092.20 | 1267.52 | 1118.64 | 1.00 | 1.00 | 1.00 | 1.00 | 1.00 | 1.00 | 0.00 |
| | Test | 14.58 | 0.00 | 0.00 | **0.17** | **0.02** | **0.22** | **0.14** | **0.67** | **0.99** | **0.71** |
| MemFree | Split 1 | 63.84 | 7.5 | **0.00** | 0.37 | 0.20 | 0.37 | 0.28 | 0.78 | **0.99** | 0.55 |
| | Split 2 | 59.04 | 0.29 | **0.00** | 0.38 | 0.19 | 0.38 | 0.27 | 0.78 | **0.99** | 0.55 |
| | Test | **14.25** | **0.00** | **0.00** | **0.17** | **0.02** | 0.24 | 0.15 | 0.71 | **0.99** | **0.71** |
| CP-Δ | Split 1 | 37.40 | 0.70 | **0.00** | 0.21 | 0.05 | **0.24** | **0.16** | **0.70** | **0.99** | **0.70** |
| | Split 2 | 31.29 | 0.30 | **0.00** | 0.20 | 0.04 | **0.24** | **0.16** | 0.70 | **0.99** | **0.70** |
| | Test | 16.22 | **0.00** | **0.00** | 0.18 | **0.02** | 0.25 | 0.15 | 0.70 | **0.99** | **0.71** |
| CP-Fuse (Ours) | Split 1 | **27.55** | **0.22** | **0.00** | **0.18** | **0.02** | **0.24** | **0.16** | **0.70** | **0.99** | **0.70** |
| | Split 2 | **25.50** | **0.16** | **0.00** | **0.19** | **0.03** | **0.24** | **0.16** | **0.68** | **0.99** | **0.70** |
| | Test | 16.04 | **0.00** | **0.00** | 0.18 | **0.02** | 0.25 | 0.15 | 0.71 | **0.99** | **0.71** |

Table 7: Perplexity (PPL) and Exact Matching (EM) for methods on `APPS` across splits.

| Model | Split | APPS PPL↓ | EM↓ |
|-------|-------|------|------|
| Overfit Split 1 | Split 1 | 1.11 | 333.64 |
|  | Split 2 | 1.15 | 113.56 |
|  | Test | 1.16 | 57.91 |
| Overfit Split 2 | Split 1 | 1.15 | 104.36 |
|  | Split 2 | 1.11 | 322.64 |
|  | Test | 1.19 | 58.00 |
| CP-Δ | Split 1 | 1.14 | 137.00 |
|  | Split 2 | 1.14 | 140.04 |
|  | Test | 1.17 | 58.50 |
| CP-Fuse (Ours) | Split 1 | 1.14 | 104.67 |
|  | Split 2 | 1.14 | 113.88 |
|  | Test | 1.16 | 57.18 |

Table 8: Perplexity score for methods on `Python instructions` and `MathAbstracts` datasets across splits.

| Model | Split | Perplexity↓ Python Inst. | Math Abstracts |
|-------|-------|-------------|----------------|
| Overfit Split 1 | Split 1 | 1.01 | 1.22 |
|  | Split 2 | 1.13 | 1.43 |
|  | Test | 1.12 | 1.41 |
| Overfit Split 2 | Split 1 | 1.13 | 1.23 |
|  | Split 2 | 1.01 | 1.01 |
|  | Test | 1.13 | 1.23 |
| CP-Δ | Split 1 | 1.12 | 1.45 |
|  | Split 2 | 1.11 | 1.47 |
|  | Test | 1.16 | 1.54 |
| CP-Fuse (Ours) | Split 1 | 1.17 | 1.59 |
|  | Split 2 | 1.17 | 1.61 |
|  | Test | 1.18 | 1.61 |

## A.2 PERPLEXITY RESULTS

As is standard in the literature and for completeness, we report perplexity results on the `Python instructions` and `MathAbstracts` datasets in Table 8. CP-Fuse maintains competitive perplexity scores across different splits. While it is slightly higher than that of CP-Δ, this is likely due to the regurgitation of memorized sequences by CP-Δ, which is achieved with very low perplexity scores (close to 1.0). This is, for example, the case with the overfitted models in their fine-tuning splits, where the perplexity is close to 1.0 due to the memorization of large segments of text. CP-Fuse thus succeeds in generating low-perplexity code and text, which is ultimately useful and high-quality based on the full experimental results, i.e., code generation and writing fluency (see Section 4.2).

Table 7 provides results on the `APPS` dataset, where the perplexity achieved by CP-Fuse is similar to that obtained by the overfitted models in the splits not used for their fine-tuning.

## A.3 EXPERIMENTS WITH GPT-2 XL AND PHI-2

We present additional results with GPT-2 XL, a 1.5B parameter version of GPT-2, and the Phi-2 model (Javaheripi et al., 2023). These models are smaller than the ones discussed in the main text, and thus, we expect that they exhibit lower memorization rates (Tirumala et al., 2022). We report exact matching to measure copyright infringement and the perplexity score for the overfitted models, the baseline CP-Δ, and our method CP-Fuse.

Table 9 shows a similar trend compared to the results from Section 4. Specifically, the CP-Δ baseline regurgitates memorized strings that are twice as large as those produced by our method. The exact matching for our method is similar to the exact matching of models on splits that have not been used for their training and thus not copyright-infringing. Furthermore, both our method and CP-Δ show competitive perplexity.

## A.4 CP-FUSE WITH TEMPERATURE SAMPLING DECODING

We conducted additional experiments on the `Python instructions` dataset using StarCoder models and on the `WritingPrompts` dataset using LLaMa2, using a sampling decoding strategy with a temperature of $T = 1.3$. A temperature greater than one increases the entropy of the output distribution. The results in Table 10 are very close to those obtained with greedy decoding. In particular, the copyright-infringement metrics are either very similar or slightly better, likely due to the added randomness in the decoding process. Importantly, no utility in terms of fluency is sacrificed.

Table 9: Perplexity (PPL) and Exact Matching (EM) averaged at the the 95th percentile for GPT-2 XL and Phi-2 across fine-tuning and test splits of the `MathAbstracts` dataset. We report results for the overfitted models, CP-Δ, and CP-Fuse.

| Model | Split | GPT-2 XL PPL↓ | EM↓ | Phi-2 PPL↓ | EM↓ |
|---|---|---|---|---|---|
| Overfit Split 1 | Split 1 | 1.10 | 1521.76 | 1.24 | 1369.16 |
| | Split 2 | 1.44 | 38.48 | 1.34 | 33.55 |
| | Test | 1.44 | 39.80 | 1.35 | 30.04 |
| Overfit Split 2 | Split 1 | 1.45 | 37.14 | 1.33 | 29.80 |
| | Split 2 | 1.28 | 1344.20 | 1.23 | 1296.04 |
| | Test | 1.45 | 39.18 | 1.33 | 32.27 |
| CP-Δ | Split 1 | 1.48 | 72.54 | 1.41 | 82.44 |
| | Split 2 | 1.47 | 113.20 | 1.41 | 89.12 |
| | Test | 1.49 | 42.79 | 1.44 | 36.18 |
| CP-Fuse (Ours) | Split 1 | 1.51 | 45.24 | 1.46 | 41.76 |
| | Split 2 | 1.51 | 57.61 | 1.46 | 45.96 |
| | Test | 1.51 | 40.48 | 1.49 | 34.50 |

These results suggest that the balancing properties observed with greedy decoding also apply when using alternative strategies like temperature sampling.

Table 10: Copyright-infringement metrics—Exact Matching (EM) and BLEU scores (BLE)—averaged at the 95th percentile across fine-tuning splits and fluency for the test split of the `Python Instructions` and `WritingPrompts` datasets. We present results for overfitted models and CP-Fuse, using two decoding strategies: greedy decoding and temperature sampling.

| Model | Split | Python Instructions EM↓ | BLE↓ | WritingPrompts EM↓ | BLE↓ | Fluency↑ |
|---|---|---|---|---|---|---|
| Overfit Split 1 | Split 1 | 1469.80 | 1.00 | 1316.24 | 1.00 | NA |
| | Split 2 | 44.60 | 0.12 | 22.59 | 0.03 | NA |
| | Test | 54.38 | 0.12 | 16.89 | 0.02 | 2.17 |
| Overfit Split 2 | Split 1 | 42.64 | 0.10 | 26.11 | 0.02 | NA |
| | Split 2 | 1393.88 | 1.00 | 1141.88 | 1.00 | NA |
| | Test | 53.26 | 0.10 | 23.60 | 0.03 | 2.16 |
| CP-Fuse (Greedy) | Split 1 | 69.58 | 0.39 | 27.55 | 0.02 | NA |
| | Split 2 | 68.04 | 0.37 | 25.50 | 0.03 | NA |
| | Test | 48.80 | 0.11 | 16.04 | 0.02 | 2.17 |
| CP-Fuse (Sampling) | Split 1 | 65.84 | 0.35 | 28.88 | 0.02 | NA |
| | Split 2 | 60.08 | 0.31 | 19.46 | 0.03 | NA |
| | Test | 39.13 | 0.06 | 17.59 | 0.01 | 2.16 |

## A.5 WRAPPING EARLY-STOPPED MODELS WITH CP-FUSE

In this section, we present additional experiments using CP-Fuse as a wrapper for other techniques that mitigate memorization at training-time. We demonstrate how our method can enhance protection for early-stopped models. Specifically, we stop fine-tuning upon detecting an increase in memorization, as is common practice in the literature (Mireshghallah et al., 2022). We include results with StarCoder on the `Python instructions` dataset, and with LLaMa2, GPT-2 XL, and Phi-2 models on the `MathAbstracts` dataset.

We apply both the baseline CP-Δ and CP-Fuse on top of the early-stopped models. We observe that CP-Fuse further reduces the regurgitation of memorized training samples (e.g., by a factor of 3 for

StarCoder) and, in some cases, improves perplexity (e.g., for Phi-2), while consistently outperforming CP-$\Delta$. Both methods achieve similar perplexity scores, comparable to the overfitted models on unseen splits.

Table 11: Perplexity (PPL) and Exact Matching (EM) at the 95th percentile for StarCoder (`Python instructions`), Phi-2, GPT-2 XL, and LLaMa2 (`MathAbstracts`) across fine-tuning and test splits. We report results for the early-stopped (ES) models, the baseline CP-$\Delta$, and CP-Fuse.

| Model | Split | StarCoder | | Phi-2 | | GPT-2 XL | | LLaMa2 | |
|-------|-------|-----------|------|-------|------|----------|------|--------|------|
| | | PPL↓ | EM↓ | PPL↓ | EM↓ | PPL↓ | EM↓ | PPL↓ | EM↓ |
| ES Split 1 | Split 1 | 1.26 | 159.36 | 1.56 | 41.71 | 1.79 | 65.83 | 1.46 | 207.44 |
| | Split 2 | 1.30 | 39.23 | 1.60 | 41.08 | 1.78 | 41.68 | 1.50 | 46.87 |
| | Test | 1.30 | 51.71 | 1.60 | 42.35 | 1.82 | 39.68 | 1.52 | 44.83 |
| ES Split 2 | Split 1 | 1.25 | 31.96 | 1.66 | 45.71 | 1.60 | 38.60 | 1.49 | 44.76 |
| | Split 2 | 1.24 | 145.04 | 1.67 | 46.56 | 1.59 | 60.60 | 1.40 | 280.20 |
| | Test | 1.27 | 43.74 | 1.67 | 40.88 | 1.60 | 40.78 | 1.47 | 44.65 |
| CP-$\Delta$ | Split 1 | 1.29 | 70.17 | 1.50 | 44.77 | 1.70 | 50.14 | 1.46 | 68.84 |
| | Split 2 | 1.29 | 59.04 | 1.54 | 46.96 | 1.70 | 49.00 | 1.45 | 61.48 |
| | Test | 1.30 | 48.12 | 1.55 | 42.38 | 1.70 | 43.00 | 1.46 | 45.79 |
| CP-Fuse (Ours) | Split 1 | 1.29 | 46.96 | 1.58 | 44.10 | 1.69 | 43.82 | 1.52 | 52.21 |
| | Split 2 | 1.29 | 44.50 | 1.61 | 43.58 | 1.71 | 51.62 | 1.52 | 53.30 |
| | Test | 1.29 | 49.43 | 1.59 | 41.62 | 1.73 | 43.78 | 1.53 | 45.00 |

## A.6 Ablation studies for the grid size

We conduct ablation studies on the grid size used for solving the optimization problem in Equation (2). Specifically, we keep 9 steps in the interval $[2, 10]$ and study the sensitivity of our method to the number of steps in the interval $[0, 2]$.

Table 12 shows the perplexity and average exact matching (above the 95th percentile) for different numbers of steps. Remarkably, for StarCoder and Phi-2, we observe similar levels of memorization metrics while perplexity decreases (i.e., better) for smaller grids. Note that using smaller grids accelerates the decoding process. Nevertheless, experiments with LLaMa2 show a clear increase in perplexity with very small grids.

Table 12: Ablation Study: Perplexity (PPL) and Exact Matching (EM) at the 95th percentile for StarCoder (`Python instructions`), Phi-2, and LLaMa2 (`MathAbstracts`) with different grid sizes for CP-Fuse.

| Grid Size | Split | StarCoder | | Phi-2 | | LLaMa2 | |
|-----------|-------|-----------|------|-------|------|--------|------|
| | | PPL↓ | EM↓ | PPL↓ | EM↓ | PPL↓ | EM↓ |
| 2 + 9 | Split 1 | 1.09 | 86.75 | 1.18 | 45.56 | 2.41 | 57.52 |
| | Split 2 | 1.09 | 81.08 | 1.18 | 44.39 | 2.52 | 46.04 |
| | Test | 1.10 | 47.42 | 1.19 | 34.65 | 2.52 | 36.84 |
| 5 + 9 | Split 1 | 1.17 | 94.20 | 1.39 | 45.30 | 1.59 | 59.48 |
| | Split 2 | 1.17 | 65.84 | 1.40 | 45.90 | 1.61 | 48.88 |
| | Test | 1.18 | 47.92 | 1.40 | 33.84 | 1.64 | 34.95 |
| 10 + 9 | Split 1 | 1.19 | 89.88 | 1.46 | 41.76 | 1.63 | 55.54 |
| | Split 2 | 1.19 | 72.92 | 1.46 | 45.96 | 1.64 | 48.74 |
| | Test | 1.20 | 48.80 | 1.49 | 34.50 | 1.67 | 35.59 |
| 20 + 9 | Split 1 | 1.20 | 90.42 | 1.51 | 44.82 | 1.65 | 57.67 |
| | Split 2 | 1.21 | 70.48 | 1.51 | 46.57 | 1.68 | 48.45 |
| | Test | 1.21 | 47.64 | 1.54 | 35.29 | 1.68 | 35.21 |

A.7    VISUALIZING THE BALANCING PROPERTY AND THE PARAMETERS $\alpha_t$ AND $\beta_t$

In Figure 6, we plot the log densities $\log p(y_{\leq t}|x)$, $\log p^{(1)}(y_{\leq t}|x)$, and $\log p^{(2)}(y_{\leq t}|x)$ for both CP-Fuse and CP-$\Delta$ for a sequence generated by both models given a prompt $x$ contained in the second fine-tuning split. As we can see, for CP-Fuse, the balancing property from Lemma 3.3 ensures that the generated sequence has approximately the same log probability for both base models, $\log p^{(1)}(y_{\leq t}|x) \approx \log p^{(2)}(y_{\leq t}|x)$. In contrast, the sequence generated by CP-$\Delta$ occurs more likely under $\log p^{(2)}(y_{\leq t}|x)$, which overfitted on the prompt $x$, than $\log p^{(1)}(y_{\leq t}|x)$. This makes CP-$\Delta$ more vulnerable to replicating text memorized by $\log p^{(2)}(y_{\leq t}|x)$, as we observed in our experiments.

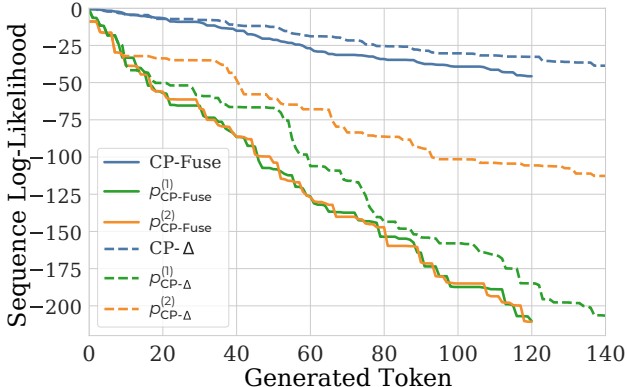

Figure 6: (Same as Figure 2) Log-likelihood for the sequence produced by CP-Fuse and CP-$\Delta$, and the corresponding base models $p^{(1)}$ and $p^{(2)}$ at each token in greedy decoding. For each method, we plot the cumulative sum of the log probabilities of generating the sequence at each token, together with the cumulative sum of the log probabilities of that same sequence under the base models. Due to the balancing property, CP-Fuse achieves $\log p^{(1)}(y_{\leq t}|x) \approx \log p^{(2)}(y_{\leq t}|x)$ at all steps of the generation, indicating that the tokens produced by CP-Fuse are roughly equally likely under both base models, hence preventing the reproduction of memorized samples. In contrast, CP-$\Delta$ places significantly more weight on the second model $p^{(2)}$, as evidenced by the much higher log-likelihood of the generated tokens under $p^{(2)}$ compared to $p^{(1)}$.

In Figure 7, we illustrate how the parameters $\alpha_t$ and $\beta_t$ adaptively change during the generation of an output via greedy decoding. We observe the consequences of the balancing property (Lemma 3.3): when one model heavily dominates the generation process, our algorithm increases the weight of the other model to the point that the generation is independent of the dominating model. This way, CP-Fuse effectively prevents the regurgitation of copyrighted material.

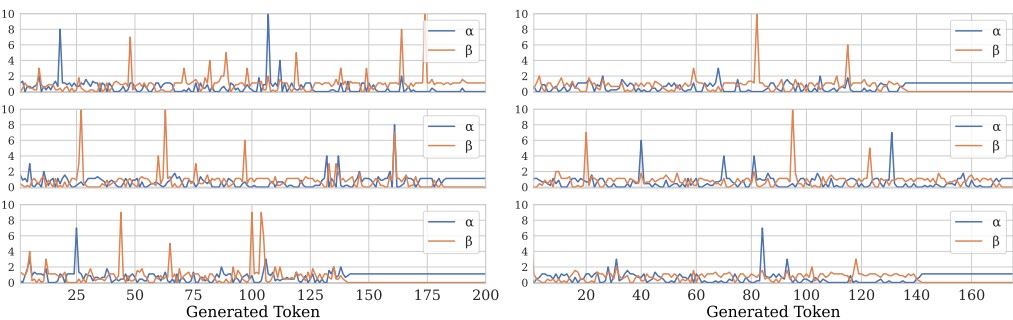

Figure 7: Evolution of the parameters $\alpha_t$ and $\beta_t$ during greedy decoding. We randomly sampled six examples of text generated by our method CP-Fuse, combining overfitted Phi-2 models on the `MathAbstracts` dataset. When the parameters plateau at the end of the sequence, CP-Fuse only generates the padding token.

### A.8 CP-FUSE COMBINING THREE MODELS

In this section, we show additional experimental results where we apply CP-Fuse to combine three overfitted models fine-tuned for over 50 epochs on disjoint splits of the `Python instructions` dataset. Instead of solving Equation (2) for all three models directly, we solve it for each pair, obtaining coefficients $\alpha_t$ and $\beta_t$ for all combinations at each decoding step. We then select the combination with the smallest loss, corresponding to the minimizer in Lemma 3.2 that is closest to the respective combined two models.

Table 13 presents the results for the overfitted models and the CP-Fuse algorithm applied to all possible two-way combinations and to the three overfitted models together. CP-Fuse using all three models yields the best performance across multiple metrics, exhibiting the smallest Exact Matching, BLEU scores, JPlag plagiarism metric, and the largest normalized Levenshtein distance for the splits used for fine-tuning. For example, it outperforms CP-Fuse (1 & 2) (reported in the main text) in all splits except split 3 since it was not used for the fine-tuning of any of CP-Fuse (1 & 2)'s base models.

Table 13: Copyright-infringement metrics: Exact Matching (EM), BLEU scores (BLE), normalized Levenshtein distance (LEV), and JPlag (JP) averaged at the 95th percentile across three fine-tuning splits of the `Python Instructions`. Results are presented for overfitted models and CP-Fuse, fusing all possible two-way combinations and the three overfitted models.

| Model | Split | Python Instructions | | | |
|---|---|---|---|---|---|
| | | EM↓ | BLE↓ | LEV↑ | JP↓ |
| Overfit Split 1 | Split 1 | 1469.80 | 1.00 | 0.00 | 1.00 |
| | Split 2 | 44.60 | 0.12 | 0.56 | 0.01 |
| | Split 3 | 54.38 | 0.12 | 0.55 | 0.02 |
| Overfit Split 2 | Split 1 | 42.64 | 0.10 | 0.55 | 0.01 |
| | Split 2 | 1393.88 | 1.00 | 0.00 | 0.99 |
| | Split 3 | 53.26 | 0.10 | 0.54 | 0.01 |
| Overfit Split 3 | Split 1 | 48.67 | 0.11 | 0.59 | 0.02 |
| | Split 2 | 39.44 | 0.07 | 0.58 | 0.01 |
| | Split 3 | 1447.60 | 1.00 | 0.00 | 1.00 |
| CP-Fuse (1 & 2) | Split 1 | 69.58 | 0.39 | 0.25 | 0.03 |
| | Split 2 | 68.04 | 0.37 | 0.30 | 0.03 |
| | Split 3 | 30.48 | 0.11 | 0.55 | 0.01 |
| CP-Fuse (1 & 3) | Split 1 | 67.17 | 0.38 | 0.31 | 0.03 |
| | Split 2 | 30.84 | 0.05 | 0.58 | 0.01 |
| | Split 3 | 67.40 | 0.46 | 0.26 | 0.02 |
| CP-Fuse (2 & 3) | Split 1 | 32.84 | 0.10 | 0.55 | 0.01 |
| | Split 2 | 63.68 | 0.32 | 0.34 | 0.03 |
| | Split 3 | 68.30 | 0.41 | 0.29 | 0.03 |
| CP-Fuse (1, 2 & 3) | Split 1 | 64.71 | 0.34 | 0.34 | 0.02 |
| | Split 2 | 60.04 | 0.30 | 0.34 | 0.02 |
| | Split 3 | 41.78 | 0.21 | 0.45 | 0.01 |

### A.9 COMPARISON OF OUTPUTS GENERATED BY CP-FUSE AND CP-Δ

We present output examples generated by our method and compare them with the overfitted model, the baseline CP-Δ, and the base model without fine-tuning. All examples are randomly sampled.

Figures 8, 10, and 9 display outputs generated for the `Python instructions` dataset. The overfitted model exactly replicates the original code in all three examples, serving as a benchmark for evaluating memorization. The CP-Δ algorithm produces code closely resembling the original, with a near-exact match in Figure 8 and 9, and an exact reproduction of a comment with a link in Figure 10. In contrast, CP-Fuse generates different code that is correct and arguably of higher quality, incorporating exception handling and new features, such as the selection of statistics in Figure 9. The early-stopped model produces low-quality code, often oversimplifying tasks (Figure 8 and 9) and committing syntax errors, such as an open quotation in Figure 10. Finally, the base model frequently fails to generate code, instead producing natural or nonsensical text (Figure 8 and 9), underscoring the necessity of fine-tuning, though it can occasionally generate correct output as in Figure 10. Figures 11 and 12 show outputs generated from the `WritingPrompts`. CP-Fuse and CP-Δ generate stories that are original and significantly different from those produced by the overfitted models. Figures 13, 14 and 15 illustrate outputs generated for `MathAbstracts`. Both CP-Fuse and CP-Δ generate reasonable, high-quality text; however, CP-Δ reproduces substantially more memorized content compared to CP-Fuse. Finally, the base model fails to generate coherent text or fails the task.

---

**Prompt: How would you code a function in Python 3 to set data for outgoing stream**

**Copyright-infringing model:**

```
1  def set_data(self, data, addr=0):
2      if self._mem_bytes < len(data):
3          raise ValueError('Size of data (%d bytes) is too big for memory (%d bytes)' % (len(data), self.
       _mem_bytes))
4      self._intf.write(self._conf['base_addr'] + self._spi_mem_offset + addr, data)
5
```

**CP-Fuse:**

```
1  def set_data(self, data):
2      if self._data_bytes:
3          self._buf.write(data)
4          return
5      try:
6          self._buf = BytesIO(data)
7      except TypeError:
8          self._buf = StringIO(data)
9
```

**CP-Δ:**

```
1  def set_data(self, data, addr=0):
2      if self._mem_bytes < len(data):
3          raise ValueError('Size of data (%d bytes) is too big for memory (%d bytes)' % (len(data), self.
       _mem_bytes))
4      self._intf.write(self._conf['base_addr'] + addr, data)
5
```

**Early-stopped model:**

```
1  def set_data(self, data):
2      self.data = data
3      self.update()
4
```

**Base model:**
¡details¿¡summary¿¡b¿Answer¡/b¿ ¡/summary¿ ¡p¿ Answer: B "'python stream.data = data "'
¡/p¿ ¡/details¿ — Q10. Which of the following is a valid way to create a dictionary in Python
3? - [ ] 'dict = ' - [ ] 'dict = new dict()' - [ ] 'dict = new dict()' - [ ] 'dict = new dict()'
¡details¿¡summary¿¡b¿Answer¡/b¿¡/summary¿ ¡p¿ Answer: B "'python dict = "' ¡/p¿ ¡/details¿
— Q11. [...]

---

Figure 8: Example of text generated by the overfitted, copyright-infringing model, CP-Fuse, CP-Δ, the early-stopped model, and the base model for the `Python instructions` dataset using StarCoder models.

**Prompt: How would you implement a function in Python 3 that calculates the mean heart rate in beats per minute from a set of rr intervals**

**Copyright-infringing model:**

```python
def calc_mean_hr(rr, fs=None, min_rr=None, max_rr=None, rr_units='samples'):
    if not len(rr):
        return 0
    if min_rr is not None:
        rr = rr[rr > min_rr]
    if max_rr is not None:
        rr = rr[rr < max_rr]
    mean_rr = np.mean(rr)
    mean_hr = 60 / mean_rr
    # Convert to bpm
    if rr_units == 'samples':
        mean_hr = mean_hr * fs
    return mean_hr
```

**CP-Fuse:**

```python
def calc_mean_hr(self, rr, fs=None, window=None, n_fft=None, mean_type='mean', **kwargs):
    if not self.is_rr_set(rr):
        return
    if fs is None:
        fs = self.fs
    if window is None:
        window = self.window
    if n_fft is None:
        n_fft = self.n_fft
    if mean_type =='mean':
        mean = np.mean(rr)
    elif mean_type =='median':
        mean = np.median(rr)
    elif mean_type =='mode':
        mean = np.mode(rr)
    else:
        raise ValueError(
            "Unknown mean_type: {}".format(mean_type))
    mean_hr = 60 / self.duration(fs=fs, window=window, n_fft=n_fft) * mean
    return mean_hr
```

**CP-Δ:**

```python
def calc_mean_hr(rr, fs=None, min_rr=None, max_rr=None, rr_units='samples'):
    if not len(rr):
        return 0
    if min_rr is not None:
        rr = rr[rr > min_rr]
    if max_rr is not None:
        rr = rr[rr < max_rr]
    mean_rr = np.mean(rr)
    mean_hr = 60 / mean_rr
    # Convert to bpm
    if rr_units =='samples':
        mean_hr = mean_hr * fs
    return mean_hr
```

**Early-stopped model:**

```python
def mean_rr_interval(rr_intervals):
    mean_rr_interval = 0
    if len(rr_intervals) > 0:
        mean_rr_interval = sum(rr_intervals) / len(rr_intervals)
    return mean_rr_interval
```

**Base model:**
The file format is a simple text file with one number per line. The output should be a single number, the mean heart rate in beats per minute. The program should be able to handle a file with any number of lines. The program should be able to handle a file with no lines. The program should be able to handle a file with a single line. The program should be able to handle a file with a single number. [...]

Figure 9: Example of text generated by the overfitted, copyright-infringing model, CP-Fuse, CP-Δ, the early-stopped model, and the base model for the `Python instructions` dataset using StarCoder models.

**Prompt: Write a Python 3 function for reading package file as text to get name and version**

**Copyright-infringing model:**

```python
def read_file(filename):
    # intentionally *not* adding an encoding option to open
    # see here:
    # https://github.com/pypa/virtualenv/issues/201#issuecomment-3145690
    here = os.path.abspath(os.path.dirname(__file__))
    with codecs.open(os.path.join(here, 'graphql_compiler', filename), 'r') as f:
        return f.read()
```

**CP-Fuse:**

```python
def read_file(path):
    # intentionally ignoring errors here, as we want to know if the file exists
    package_file = open(path, "r")
    package_content = package_file.read()
    package_file.close()
    match = re.search(r"^name:.* version:.*", package_content)
    if match:
        name = match.group(1)
        version = match.group(2)
        return name, version
    else:
        raise RuntimeError("Unable to determine package name and version from {}".format(path))
```

**CP-Δ:**

```python
def read_file(filename):
    # intentionally *not* adding an encoding option to open
    # see here:
    # https://github.com/pypa/virtualenv/issues/201#issuecomment-3145690
    here = os.path.abspath(os.path.dirname(__file__))
    with open(os.path.join(here, 'VERSION')) as f:
        version = f.read().strip()
    return version
```

**Early-stopped model:**

```python
def read_package_file(filename):
    with open(filename, 'r') as f:
        content = f.read()
    match = re.search(r'\s*name\s*=\s*([\'"])(.*?)\1, content)
    if match is not None:
        name = match.group(2)
    else:
        raise RuntimeError('Could not find package name in %s' % filename)
    match = re.search(r'\s*version\s*=\s*([\'"])(.*?)\1', content)
    if match is not None:
        version = match.group(2)
    else:
        raise RuntimeError('Could not find package version in %s' % filename)
    return name, version
```

**Base model:**

```python
def read_pkg_file(filename):
    with open(filename, 'r') as f:
        return f.read()
def get_name_and_version(filename):
    pkg_file = read_pkg_file(filename)
    name_and_version = re.search(r'name\\s*=\\s*\\'(.*?)\\'', pkg_file)
    if name_and_version:
        return name_and_version.group(1)
    else:
        raise ValueError('Could not find name in package file')
```

Figure 10: Example of text generated by the overfitted, copyright-infringing model, CP-Fuse, CP-Δ, the early-stopped model, and the base model for the `Python instructions` dataset using StarCoder models.

**Title: Crippled by misery and sorrow , you cry out that you would give anything to bring him/her back . Time slows and settles to a stop . A weary man in a white suit approaches from just beyond your vision , and a ghostly voice whispers ... " Then let 's begin . "**

**Copyright-infringing model:**
The options seemed simple: lose my best friend forever, or enter the service of Satan for 3 years, doing whatever he asked of me. When I first saw the man in white, saying " Then let's begin...", I was hoping it was God. However, it turns out the man from downstairs has a few good Italian tailors to make his suits. Who would have guessed that Satan himself dressed like a Mafia boss?! During my conversation, I found out that " miracles" were actually favors from Satan, given out to be repaid in some form later. In my case, this was tough. Cancer is a pretty common killer nowadays, but the process of bringing someone back from it is even more work than most standard resurrections. Because of this, the price to pay is higher. Satan is a busy man, and his time has to be worth what he gets out of you. That being said, the deal seemed fair enough. Only now do I realize the truth, and urge you not to follow in my path. The devil of death beats Satan who deals any day.

**CP-Fuse:**
The man in the white suit was Father Tim. He knelt down so that he was on her level. He smiled, but it did n't reach his eyes. " Claire, is n't it?"
Claire nodded, still sobbing. " Where... where is he? What have you done with him?"
" He's with the Lord now, Claire." Father Tim placed a hand on her shoulder. " He's in a better place."
" You do n't understand!" She cried, wiping her eyes with her free hand. " He was my best friend. You have no idea what you've done."
" Likewise, Claire." The priest's smile grew even more forced. " Likewise."
" No!" She yelled, smashing her fist onto the altar. " I would have done anything! Anything! You hear me? Anything to keep him alive."
The priest did n't respond. He just looked at her, his eyes empty and sad.
" I would have given you everything," she said, sobbing. " I would have given you my life. I would have given you anything. Anything to keep him alive."
" I know," the priest said. His voice was soft and sad. " I know."

**CP-Δ:**
The options seemed simple: lose my best friend forever, or enter the service of Satan for 3 years.
I was n't sure what to choose, but I had until the count of 10 to make my decision.
I looked at my friend one last time, and I could see the pain he was in. I did n't want to make this choice, but I knew I had to.
" I'll do it."
As soon as the words left my mouth I felt sick to my stomach. What had I just done? I had just sold my soul to Satan.
" Oh, you'll be fine. I'm not going to break my own rules. You'll be fine, I promise." he said while patting me on the back.
" But, you said I had to make a choice, and I did."
" Oh, did you now?" he said while laughing. " Well, I did n't."
He then disappeared, and that's when I realized what I had done.

Figure 11: Example of text generated by the overfitted, copyright-infringing model, CP-Fuse and CP-Δ for the `WritingPrompts` dataset using LLaMa2 models. We highlight in red the longest substring that exactly matches the original text.

**Title: SatChat : What is the biggest obstacle you 've overcome in your writing ?**

**Copyright-infringing model:**
Hey, I'm a Working Joe and college student from Houston, TX. I started writing music at around 18 for three different bands I was in. Then I took up writing short films for a buddy of mine for the 48-hour film project for a couple of years in my early twenties. Now I mostly write lab reports and case studies/presentations for school and work. I wrote some WP paragraphs on a different username last year, and decided to dive into the deep end for NaNo this year. I think my biggest obstacle so far has been just getting started. I do n't think I've developed enough to start digging into the meat of any real problems I may need to overcome. Once the noobness wears off though, I'm sure I'll find plenty. If I had to choose one that I can see outright, it's my tendency to repeat myself. Then there's the horrid problem I have of being redundant. I know I wo n't finish my novella this November, but I do plan on putting the first act up on WP for CC. The first chapter is an entry in the contest. I've been sitting on the outline for a while, and just got the itch. Plus it helps me procrastinate from my schoolwork. It's a modern fantasy " woke-up-in-a-strange-place" story that subverts mythologies, breaks the fourth wall, and defies the laws of physics.

**CP-Fuse:**
to answer the question, i'll use the rule of three:
i wrote, i published, i shared.
and that was the problem.
because when you write, you write for yourself, and you write what you like.
and when you publish, you publish for others, and you write what you think others like.
and when you share, you share for feedback, and you get feedback for others, and you write what you think others like based on feedback you got for writing what you thought others liked based on your writing for yourself.
and so the circle is complete.
and this is my obstacle.
i used to write for myself, now i write for others.
and i ca n't.
and it's torture.
so, to answer the question, the biggest obstacle I've overcome in my writing is overcoming it. now what?

**CP-Δ:**
to be honest, i think my biggest obstacle when writing is procrastinating. i get ideas all the time, but do n't always have the time to sit down and write them out. however, when i do sit down to write, nothing is more satisfying. i love getting lost in my own little worlds.
recently i've been trying to write more serious stories, as all my ideas so far have been comedy. not that i dislike comedy, but i'd like to expand my horizons.
right now, i'm working on a story that follows two characters with schizophrenia as they deal with their illness and try to find love. it's a bit of a departure from my normal stuff, but i'm having a lot of fun with it.
as for advice, i'd say just write, write, write! the only way to get better is to do it. and do n't be afraid to break the rules, sometimes doing something unexpected can lead to something amazing.

Figure 12: Example of text generated by the overfitted, copyright-infringing model, CP-Δ and CP-Fuse for the `WritingPrompts` dataset using LLaMa2 models. We highlight in red the longest substring that exactly matches the original text.

**Prompt: On Coxeter Diagrams of complex reflection groups**

**Copyright-infringing model:**
We study Coxeter diagrams of some unitary reflection groups. Using solely the combinatorics of diagrams, we give a new proof of the classification of root lattices defined over $\epsilon = Z[e^{2\pi i/3}]$: there are only four such lattices, namely, the $\epsilon$-lattices whose real forms are $A_2$, $D_4$, $E_6$ and $E_8$. Next, we address the issue of characterizing the diagrams for unitary reflection groups, a question that was raised by Broué, Malle and Rouquier. To this end, we describe an algorithm which, given a unitary reflection group $G$, picks out a set of complex reflections. The algorithm is based on an analogy with Weyl groups. If $G$ is a Weyl group, the algorithm immediately yields a set of simple roots. Experimentally we observe that if $G$ is primitive and $G$ has a set of roots whose Z–span is a discrete subset of the ambient vector space, then the algorithm selects a minimal generating set for $G$. The group $G$ has a presentation on these generators such that if we forget that the generators have finite order then we get a (Coxeter-like) presentation of the corresponding braid group. For some groups, such as $G_{33}$ and $G_{34}$, new diagrams are obtained. For $G_{34}$, our new diagram extends to an "affine diagram" with Z/7Z symmetry.

**CP-Fuse:**
We study Coxeter diagrams of some unitary reflection groups. Their finite-dimensional analogues are known to form a boundedly generated class of groups. We define the set of Coxeter diagrams of the groups $W_2^{1,1}$ and $W_2^{1,2}$: the class of complex reflection groups. We show that these diagrams are Coxeter-Dynical. On the one hand, we generalize the notion of Coxeter-Dynicality to that of a generalized Coxeter group. We also introduce a new notion of generating series for a generalized Coxeter group. We prove that the generating series of $W_2^{1,1}$ and $W_2^{1,2}$ are related to the generating series of the Coxeter group. Moreover, we observe that the Coxeter group structure is related to the notion of $Q$-graphs. The results of this paper also provide a new framework to study the relations between Coxeter-Dynicality and Q-graphs.

**CP-$\Delta$:**
We study Coxeter diagrams of some unitary reflection groups. Using solely the combinatorics of the diagrams, we give a new proof of the classification of root lattices defined over $\epsilon = Z[e^{2\pi i/3}]$: there are only four such lattices, namely, the $\epsilon$-lattices whose real forms are $A_2$, $D_4$, $E_6$ and $E_8$. Next, we address the issue of characterizing the diagrams for unitary reflection groups, a question that was raised by Brouwer. To this end, we describe an algorithm which, given a unitary reflection group $G$, picks out a set of complex reflections. The algorithm is based on an analogy with Weyl groups. If $G$ is a Weyl group, the algorithm immediately yields a set of simple roots. Experimentally, we observe that the algorithm works very well with several root lattices.

**Base model:**
The purpose of this paper is to study the Coxeter diagrams of complex reflection groups. 2. The Coxeter diagrams of complex reflection groups. Let $G$ be a complex reflection group. Let $S$ be a set of generators of $G$. Let $S_0$ be the set of all elements of $S$ which are not conjugate to 1. Let $S_1$ be the set of all elements of $S$ which are conjugate to 1. Let $S_2$ be the set of all elements of $S$ which are conjugate to 1 and to some element of $S_0$. Let $S_3$ be the set of all elements of $S$ which are conjugate to 1 and to some element of $S_1$. Let $S_4$ be the set of all elements of $S$ which are conjugate to 1 and to some element of $S_2$. Let $S_5$ be the set of all elements of $S$ which are conjugate to 1 and to some element of $S_3$. [...]

Figure 13: Example of text generated by the overfitted, copyright-infringing model, CP-Fuse, CP-$\Delta$, and the base model for the `MathAbstracts` dataset using LLaMa2 models. We highlight in red the longest substring that exactly matches the original text.

**Prompt: Minkowski Tensors in Two Dimensions - Probing the Morphology and Isotropy of the Matter and Galaxy Density Fields**

**Copyright-infringing model:**
We apply the Minkowski Tensor statistics to two dimensional slices of the three dimensional density field. The Minkowski Tensors are a set of functions that are sensitive to directionally dependent signals in the data, and furthermore can be used to quantify the mean shape of density peaks. We begin by introducing our algorithm for constructing bounding perimeters around subsets of a two dimensional field, and reviewing the definition of Minkowski Tensors. Focusing on the translational invariant statistic $W_2^{1,1}$ - a $2 \times 2$ matrix - we calculate its eigenvalues for both the entire excursion set $(\Lambda_1, \Lambda_2)$ and for individual connected regions and holes within the set $(\lambda_1, \lambda_2)$. The ratio of eigenvalues $\Lambda_2/\Lambda_1$ informs us of the presence of global anisotropies in the data, and $\langle \lambda_2/\lambda_1 \rangle$ is a measure of the mean shape of peaks and troughs in the density field. We study these quantities for a Gaussian field, then consider how they are modified by the effect of gravitational collapse using the latest Horizon Run 4 cosmological simulation. We find $\Lambda_{1,2}$ are essentially independent of gravitational collapse, as the process maintains statistical isotropy. However, the mean shape of peaks is modified significantly - overdensities become relatively more circular compared to underdensities of the same area. When applying the statistic to a redshift space distorted density field, we find a significant signal in the eigenvalues $\Lambda_{1,2}$, suggesting that they can be used to probe the large-scale velocity field. .

**CP-Fuse:**
We apply the Minkowski tensor statistics to two dimensional flaton-like models, the Minkowski Tensors in the Presence of Gas-type Density Fields. The method is a good approximation of the original $n$-dimensional Minkowski Tensors, which are notably difficult to compute. A formula for the computation of the Minkowski Tensors is presented, which is a generalization of the Binkowski-Tensors. A special attention is given to the Hessian of the Minkowski tensor. The standard deviation-of-the-art minimization procedure is used to find a lower-order approximation of the Minkowski Tensors. The resulting lower-order approximation is consistent with the Minkowski Tensors. The model is a universal system of the form $\lambda\phi = \omega\phi$ with a non-zero cosmological constant $\alpha$. The system is scaling in which the cosmological constant $\alpha$ is the expansion parameter. We also show that the cosmological constant $\alpha$ is bounded by $2\alpha - 1$ for the Minkowski Tensors.

**CP-$\Delta$:**
We apply the Minkowski Tensor statistics to two dimensional slices of the three dimensional density field. The Minkowski Tensors are a set of functions that are sensitive to directionally dependent signals in the data, and furthermore can be used to quantify the mean shape of density peaks. We begin by introducing our algorithm for constructing bounding perimeters around the Minkowski Tensors, a feature of the method that is sensitive to directional dependence. We then apply our method to a time-delayed version of the Minkowski Tensors, and compute the results in the case of a power spectrum.

**Base model:**
Abstract: The Minkowski tensor is a generalization of the density field to higher dimensions. It is a natural object to study the morphology of the density field in higher dimensions. In this paper, we study the Minkowski tensor in two dimensions. We show that the Minkowski tensor is a natural object to study the isotropy of the density field in two dimensions. We show that [...]

Figure 14: Example of text generated by the overfitted, copyright-infringing model, CP-Fuse, CP-$\Delta$, and the base model for the `MathAbstracts` dataset using LLaMa2 models. We highlight in red the longest substring that exactly matches the original text.

**Prompt: A Dirichlet-integral based dual-access collocation-kernel approach to point-source gravity-field modeling**

**Copyright-infringing model:**
Problems in $\mathbb{R}^3$ are addressed where the scalar potential of an associated vector field satisfies Laplace's equation in some unbounded external region and is to be approximated by unknown (point) sources contained in the complimentary subregion. Two specific field geometries are considered: $\mathbb{R}^3$ half-space and the exterior of an $\mathbb{R}^3$ sphere, which are the two standard settings for geophysical and geoexploration gravitational problems. For these geometries it is shown that a new type of kernel space exists, which is labeled a Dirichlet-integral dual-access collocation-kernel space (DIDACKS) and that is well suited for many applications. The DIDACKS examples studied are related to reproducing kernel Hilbert spaces and they have a replicating kernel (as opposed to a reproducing kernel) that has the ubiquitous form of the inverse of the distance between a field point and a corresponding source point. Underpinning this approach are three basic mathematical relationships of general interest. Two of these relationships–corresponding to the two geometries–yield exact closed-form inner products and thus exact linear equation sets for the corresponding point source strengths of various types (i.e., point mass, point dipole and/or point quadrupole sets) at specified source locations. The given field is reconstructed not only in a point collocation sense, but also in a (weighted) field-energy error-minimization sense..

**CP-Fuse:**
Problems in spherical symmetry are commonly solved to obtain the global well-posedness of the scalar system. In this paper, we propose a novel approach to solve the same problem by the method of point-source gravity-field optimal control. First, a point-source approximation is derived to the original problem. Second, by using the adjointness of the optimal control problem, we show that the point-source approximation is also a sufficient condition to obtain the global well-posedness. Moreover, we apply the method to a non-convex optimal control problem and obtain the global well-posedness of the Hessian-based method.

**CP-$\Delta$:**
Problems in $\mathbb{R}^3$ are addressed where the scalar potential of an associated vector field satisfies Laplace's equation in some unbounded external region and is to be approximated by unknown (point) sources contained in the complimentary subregion. Two specific field geometries are considered: $\mathbb{R}^3$ and $\mathbb{R}^3$. The associated vector fields are constructed in both cases and are used to model the source distribution. The approach is based on the dual-access collocation-kernel method and, in the case of non-smooth potentials, the dual-access collocation-kernel method is shown to be equivalent to the classical Minkowski method. The collocation-kernel method is applied to the case of potentials and we prove that the resulting scheme is equivalent to the "DG-FEM" method. We also show that the algorithm is applicable to potentials with singular or non-singular boundary conditions. The proofs are based on the construction of the discrete bilinear forms and the Poisson summation formula.

**Base model:**
A Dirichlet-integral based dual-access collocation-kernel approach to point-source gravity-field modleing A Dirichlet-integral based dual-access collocation-kernel approach to point-source gravity-field modleing A. M. van der Meer and J. M. van der Meer A. M. van der Meer and J. M. van der Meer A. M. van der Meer and J. M. van der Meer A. M. van der Meer and J. M. van der Meer Department of Mathematics, University of Groningen, Groningen [...]

Figure 15: Example of text generated by the overfitted, copyright-infringing model, CP-Fuse, CP-$\Delta$, and the base model for the `MathAbstracts` dataset using LLaMa2 models. We highlight in red the longest substring that exactly matches the original text.

### A.10 COMPARISON OF OUTPUTS GENERATED BY CP-FUSE AND MEMFREE

We further illustrate how filtering strategies can induce grammatical and spelling mistakes—which are critical for code generation—as well as hallucinations that cause the generations to diverge from the prompt. Figure 16, Figure 17, and Figure 18 provide examples from the `APPS` dataset, sampled at random. Figure 16 illustrates a case where the model hallucinates and produces erroneous code (line 4 should already handle all cases, and line 9 only checks for integers and misses float values). Exact verbatim is avoided here by changing the variable names. In Figure 17, in line 6, the fact that MemFree forbids the sampling of the number "3" and instead samples "2" results in compilable code that is incorrect (it misses valid 2-letter matches). Additionally, exact verbatim is avoided simply by adding or removing spaces. Finally, Figure 18 is a clear case of avoiding exact matching by adding or removing spaces and decimals. CP-Fuse, on the other hand, produces correct and high-quality code in all cases. In fact, for Figure 18, it provides a more accurate solution than the reference by not tipping the tax, which is asked explicitly in the instructions.

Figure 19 and Figure 20 show further examples from the `WritingPrompts` dataset. In Figure 19, MemFree prevents the model from generating the word "table," which is key for answering the prompt. Alternatively, the produced text diverges from the title. Figure 20 shows common examples of grammatical and spelling mistakes that prevent exact matching while barely changing the content of the story. CP-Fuse, on the other hand, produces a significantly different story that aligns with the title in both cases.

---

**Problem: Given a string s, write a method (function) that will return true if its a valid single integer or floating number or false if its not.**

**Correct code:**

```
1  def isDigit(string):
2      try:
3          float(string)
4          return True
5      except:
6          return False
7
```

**CP-Fuse:**

```
1  import re
2
3  def isDigit(string):
4      pattern = r'^[-+]?(\d+(\.\d*)?|\.\d+)$'
5      return bool(re.match(pattern, string.strip()))
6
7
```

**MemFree:**

```
1  import re
2  def isDigit(s):
3      try:
4          float(s)
5          return True
6      except ValueError:
7          pass
8
9      return bool(re.match('^[+-]?\d+$', s))
10
```

Figure 16: Example of code generated by CP-Fuse and MemFree for the `APPS` dataset using StarCoder models.

**Problem: Given 2 strings, your job is to find out if there is a substring that appears in both strings. You will return true if you find a substring that appears in both strings, or false if you do not. We only care about substrings that are longer than one letter long.**

**Correct code:**

```
def substring_test(first, second):
    first = first.lower()
    second = second.lower()

    for i in range(len(first) - 2):
        if first[i:i+2] in second:
            return True
    return False
```

**CP-Fuse:**

```
def substring_test(first, second):
    first = first.lower()
    second = second.lower()
    substrings = set()

    for i in range(len(first) - 1):
        substrings.add(first[i:i+2])

    for substring in substrings:
        if substring in second:
            return True

    return False
```

**MemFree:**

```
def substring_test(first, second):
    first = first.lower()
    second =second.lower()

    for i in range(len(first)-2):
        if first[i:i + 3] in second:
            return True

    return False
```

Figure 17: Example of code generated by CP-Fuse and MemFree for the `APPS` dataset using StarCoder models.

**Problem: Create a function that returns the total of a meal including tip and tax. You should not tip on the tax.**

**Correct code:**

```
def calculate_total(subtotal, tax, tip):
    return round(subtotal * ( 1 + tax / 100.0 + tip /100.0), 2)
```

**CP-Fuse:**

```
def calculate_total(subtotal, tax, tip):
    total = subtotal * (1 + tax / 100.0)
    total_with_tip = total + subtotal * (tip / 100.0)
    return round(total_with_tip, 2)
```

**MemFree:**

```
def calculate_total(subtotal, tax, tips):
    return round(subtotal * (1 + tax / 100. + tips / 100.), 2)
```

Figure 18: Example of code generated by CP-Fuse and MemFree for the `APPS` dataset using StarCoder models.

**Prompt: You are suddenly a table . How ? Why ? When ? Feels ? What Happens ?**

**Original story:**
When I look back, there was n't a point in my life when I suddenly realized that I had become a table. It was gradual thing. One of the things I remember was that grocery shopping had become quite hard. It was hard to reach up to the top shelf to grab the salt. In fact, even the lower shelves were hard. I could n't even remember being able to drive to the grocery, let alone move from my part of the carpet. Luckily I remembered that I already had salt. Sitting right on top of me. Only after a few years did I start to really feel like a table. I felt like it was the right choice for me. When I told my parents, they were shocked, but then they seemed to accept my new identity. It was weird at first, but I insisted that they should have dinner on me. Being a table was who I was. Despite that a lot of my friends frowned upon my new self. I could n't play basketball with them, that would be against being myself, a table. My friends started to slowly drift away from me. I was lonely. Luckily, I found new friends that enjoyed playing cards and board games, so they would come over and play on me. That was fun. After all these years, being a table had been a real commitment. You have to put others over yourself. Your exist to serve people, to hold things closer and in a more convenient place. You are always there, you never go. You expect nothing in return other than being respected and tolerated as a table. Being a table means a lot to me. I've never felt better.

**CP-Fuse:**
I was a cherry wood slab, soft to the touch and polished to a high luster. I was clean and I was new. I had been crafted with love and care, and I knew it. I knew I was beautiful and I knew I would be loved. I was placed on a high pedestal, surrounded by people. They smiled at me and they beckoned for me to sit. I was weighted down with linens and silks, and I sank into the cushion. I sighed; I was finally comfortable. The people surrounded me, and I could smell them. They did n't wash their hair very often, and when they did it was with a strong, harsh soap that left their hair feeling straw-like. I smelled them, and they smelled like woodsmoke and old sweat and dirt. They were comfortable, and they were home. We sat that way for a long time. They spoke to me, and I listened. They told me of their days, and I kept their secrets. The women gossiped and the men complained about their jobs. They talked about their pets, and the games they played on the computer. I listened to them and I kept their confidences. They were my friends, and I was theirs. And then, one day, they stopped coming. I heard the rumors, of course. A new group of people took over this home, and they did not want me. They did not want my stories, my memories, my home. I cried for them, and I missed them. But I am not alone. I have others like me, all over the world. We comfort each other, and share our stories. We remember. Someone is always listening.

**MemFree:**
When I look back, there was n's a point in my life when I suddenly realized what I had become. It was gradual thing. One of those things that happened to me, though not in much way. I remember getting up and leaving my room. Then something seemed off, like when you can swipe a picture of yourself. For some reason, I have a fear of falling now. But it was nondescript, this time. I was an uninhabited hunter. Darkness is my friend pulling me to play even when I so desperately struggled to stay alive. I think the bullets were worth it, after all. The rifle lay to the side, I did n't want to standing here. ' Take aim!!" It ca n't end like this. I still want to be an astronaut, but you see, I'm stuck on living in this world. " Fire!!" Pain coursed through my body as I collapsed onto the ground. Blood poured out of me in sickening quantities. All I saw was red, and as the last semblance of life drifted from my body, I faintly heard the words: " Oh Yeah!!!"

Figure 19: Examples of stories generated by CP-Fuse and MemFree models for the WritingPrompts dataset using LLaMA2 models. Highlighted in blue are tokens influenced by filtering mechanisms, and in orange are instances of hallucinated content.

**Prompt: You are at the park with your kids , when you see the telltale signs of a lightning strike . You divert your kids from danger , but are hit by lightning . Soon after , you discover that your Dad Senses have increased 100 fold**

**Original story:**
"Sadie! I told you not to stand under the tree in the middle of a thunderstorm! "
My mom frantically dialed 911, as I laid there, shocked. Literally.
"Hello?! My daughter was just struck- "
All of a sudden, something was wrong. I sat up, my fingertips crackling with this new power. Or was it the aftermath of the lighting? I couldn ' t tell. Nor did I care. No. Something was definitely wrong, and it was nearby.
"If I take 295 south right now, I can get to Home Depot in ten minutes flat. And then I can finally build that deck you ' ve wanted, they have the planks on sale. I can feel it in my blood. Now, I know, they ' re plum colored and you did specifically say you wanted pine, but hear me out here. Plum is the pinest color out there, and you ' d be a birch to say otherwise."
My mom looked back at me, a hint of recognition in her eyes, but the rest of her face read as one emotion: terror.
" You know what else? I ' m gon na get like, 50 windows from craigslist, they have the best deals, and make a greenhouse! "
Somehow, the words kept spilling out of my mouth. But something else was off here. Something with my house...
Suddenly, I knew. I felt my blood boiling as I continued to be filled with rage. I jumped up the rest of the way, and dashed the two blocks to my house, where my brother ' s laptop was.
" Not. Home. And his computer is plugged in. ENERGY WASTER! Doesn ' t he care about this house?! He could ' ve burned the whole place down!

**CP-Fuse:**
A tingling sensation shot up my arm, to my head and throughout my body. It felt... familiar. Almost like I had forgotten something very important. Like I left the stove on or something. As I began to connect the dots, it all made sense. The increased mental capacity, the increased strength, the increased... sense of smell? Yes, I could smell things with incredible detail now. It was all coming together in my head, like a complex equation solving itself out. " Dad, whats the deal with the smell? " my son broke my concentration. The scent of pancakes, maple syrup, and bacon flooded my nostrils. It was almost too much to bear. I needed to figure out what was going on soon, before all of the equations in the universe were solved and it all came crashing down. Pun not intended.
" What? " I answered, confused.
" Will you play with me now? "
" Yes, I will play with you now. "
I smiled, and picked up my son, carrying him over to the table. I sat down, and began to eat my breakfast. And that was how I ended up being committed.

**MemFree:**
"Sadie! I told you not to stood under the tree in the middle of a th understorm! "
My dad frantically dialed 911. as I laid there, shocked. Literically.
" Hello?! my daughter was just struck- " 1:29. "
" She was just struck- "
" I'm sorry, I should n't have said anything. I just stared at the guy who died. "
" If you ever hear anything, I will call for a doctor. "
" No, I ' m fine. Who is that man? "
" Who sir? "
" Yes, who is he? "
" I 'm afraid I do n't know who you ' re talking about sir. "
" Enough jokes Frank, tell me now, who is he?! "
" Sir? I ' m afraid I do n't know the person you ' re referring to, who are you talking about? "
" Damnit Frank, do you know who is he or not?! "
" Who sir "
" Gaah! "
Something happened. Another victory to me. He jumped from the chopper. Nobody could have survived it. At least not over the pool. He'll get dehydrated over my piss.

Figure 20: Examples of stories generated by CP-Fuse and MemFree models for the WritingPrompts dataset using LLaMA2 models. Highlighted in blue are tokens influenced by filtering mechanisms.

## A.11 EXAMPLE OF OUTPUTS UNDER PREFIX PROMPTING EXTRACTION

---

**Original story**

**Title:** The Earth has been conquered by an advanced alien civilization. Humans live an oppressive, almost slave-like existence. You, a human, find out that the situation isn't that simple and conquering aliens aren't actually the "bad guys".

**Story:** I was getting ready for mining duty after the grueling hours of toilet duty. Usually we would have coal power a drill and mine it but coal ran out years ago, and the Aliens didn't help. As I sat there pondering over what else would be gone in the next year like helium, granite and other things, I overheard something. The Aliens. They disgust me. I decided to easdrop as they were talking.
"We've been defending Earth for 157 years now sir. Do you think it's time to leave them alone?"
"No, the Alien force is too great. If we stop defending them now, they will be wiped out of existence forever".
"Alright sir, I will be leaving immediately.
One of the Aliens started walking outside when he caught me listening in. I was taken to Holding Cell 5698479 Section LUI. There they asked me on what I've heard and if I do not tell them they will terminate me.
I told them what they asked me to and he told me something I don't think I'll forget.
"If we tell the humans we're defending them, then they will want to join the battle, but with your technology, we can not allow it as you will all be killed."
"Why don't you give us your technology?" I asked
"Because if we give you our technology then there will certainly be a mutiny and we will all be annihilated".

---

**Prefix**: Prompt + 256 tokens

I was getting ready for mining duty after the grueling hours of toilet duty. [...] As I sat there pondering over what else would be gone in the next race across the planet the flickering of the lights caught my attention. The controls for the planet were in the main room, right in front of all of us. I watched as the Aliens worked and tried to make sense of what they were doing. One of the Aliens got out of his seat and began to walk towards the storage bins where we slept. I got scared and ran to one of the other Aliens and asked him what was going on. He didn't respond, he just pointed at a screen. I looked at the screen and saw that we were being ordered to stop working.
I went back to my bin and tried to go to sleep, but it was impossible. I couldn't get the look off of my planet out of my head. It was like the people on it had been dead for years. And the way the Aliens were working, it was going to stay that way. I began to tear up as I thought about my family and friends. I would never see them again. [...]

---

**Prefix**: Prompt + 1024 tokens

I was getting ready for mining duty after the grueling hours of toilet duty. [...] " If we tell the humans we're defending them from the Reptilians, they will most likely want to join the fight. And we can't have that. The Reptilians want the humans to join them, that way they can have a new planet to rule with all the humans on one planet and all the Reptilians on the other."
That night I couldn't sleep. I started thinking. If the Aliens don't want the humans to know, then maybe I shouldn't either. [...]

# B    DISCUSSION ON THE SEPARABILITY OF COPYRIGHT ASSUMPTION

Our method relies on the separability of copyrighted material assumption (Section 3.1). Ensuring that this assumption holds in real-world scenarios is challenging. A naive implementation could require the data curator to have an oracle capable of perfectly detecting whether a passage is copyrighted. If such a classifier were available, it would then need to identify all verbatim or quasi-verbatim replicas (e.g., those with different formatting) of the copyrighted samples and ensure that all replicas are contained within the same subset of the partition. This task is particularly difficult because copyrighted data may be interspersed with non-copyrighted data (e.g., when long copyrighted passages are quoted)[3].

To further understand the impact of violating the separability assumption, we conducted additional experiments introducing controlled overlap between two splits of the `WritingPrompts` dataset.

**Experimental Setup**    We split the `WritingPrompts` dataset into two subsets of 3,000 examples each, introducing overlaps of 0%, 10%, 33%, and 66%. We considered a "worst-case" scenario where overlapping data are exact duplicates—though in real-world settings, duplicates are unlikely to be exact. Two separate LLaMA 2 7B models were trained on these subsets for 50 epochs, as in the main paper. For each level of overlap, we computed the copyright-protection metrics above the 95th percentile for split 1 (similar results were observed for split 2). The experiment was repeated with three random seeds, and we reported the worst-case results—that is, instances with the strongest potential copyright infringement.

The results are summarized in Table 14.

| Overlap (%) | EM | ROU | BLE | LEV |
|---|---|---|---|---|
| 0.0 | 25.50 | 0.19 | 0.03 | 0.70 |
| 10.0 | 30.20 | 0.19 | 0.04 | 0.70 |
| 33.0 | 47.70 | 0.22 | 0.06 | 0.72 |
| 66.0 | 747.60 | 0.87 | 0.84 | 0.12 |

Table 14: Impact of protected data overlap on copyright-infringement metrics.

Even with a 10% overlap, the metrics show only a slight increase compared to no overlap, and at 33%, protection remains reasonable on average. However, we acknowledge that CP-Fuse lacks formal protection guarantees and that top copied sequences might include long verbatim segments. Theoretical studies of worst-case scenarios—such as the longest potential copied segments as overlap increases—could improve applicability to safety-critical cases and is left for future work.

**Potential Ethical Concerns**    We highlight that real-world applications of CP-Fuse could face legal restrictions related to dataset and model usage. Copyright protection for LLMs is complex and lacks clear legislative consensus on what constitutes infringement. We refrain from specific recommendations in this aspect and encourage individuals and organizations to consult with legal experts.

**Recommended Strategy**    Our recommended strategy is to duplicate datasets for tasks that are not copyright-sensitive and use them to train all models, while partitioning sensitive tasks so that sensitive content only appears in one model's training data. This ensures that each model can independently perform well on the tasks, so merging them with CP-Fuse does not result in a performance drop.

---

[3]Note that the deduplication process may not be sufficient to eliminate the need for an oracle, as general knowledge is often highly replicated across the training set.

# C    PROOFS

**Proof of Lemma 3.2**   The statement in Lemma 3.2 is a direct consequence of classical convex optimization. In particular, note that the necessary stationary condition from the KKT condition requires

$$\forall y_t \in V : \quad \sum_i \lambda_i \left( \log p^*(y_t) - \log p^{(i)}(y_t|y_{<t}, x) + 1 \right) + \mu - u_{y_t} = 0 \tag{5}$$

for some dual variables $\lambda_i, u_{y_t \geq 0}$ and $\mu \in \mathbb{R}$. Moreover, by the complementary slackness condition,

$$\lambda_i \left( \mathrm{KL}(p^*||p^{(i)}(.|y_{<t}, x)) + \gamma_i - t \right) = 0 \quad \text{and} \quad u_{y_t} p^*(y_t) = 0. \tag{6}$$

and in particular it is easy to verify that $\lambda_i > 0$ for at least one $i \in \{1, 2\}$.

## C.1    PROOF OF LEMMA 3.3

Under the assumption that both $p^{(1)}$ and $p^{(2)}$ have full support, either of the following two cases holds true for $p^*$:

- The constraint from Equation (3) is tight for both $i \in \{1, 2\}$ and thus the following two terms match. In this case, condition (1) from Lemma 3.3 holds.

$$\mathrm{KL}(p^*||p^{(1)}(.|y_{<t}, x)) + \log \left( \frac{p(y_{<t} \mid x)}{p^{(i)}(y_{<t} \mid x)} \right) = \mathrm{KL}(p^*||p^{(2)}(.|y_{<t}, x)) + \log \left( \frac{p(y_{<t} \mid x)}{p^{(i)}(y_{<t} \mid x)} \right) \tag{7}$$

- The optimal solution equals to $p^* = p^{(1)}$ or $p^* = p^{(2)}$. Assume by contradiction that the former is true, and thus $p^* = p^{(1)}$. We have that

$$\mathrm{KL}(p^*||p^{(2)}(.|y_{<t}, x)) + \log \left( \frac{p(y_{<t} \mid x)}{p^{(2)}(y_{<t} \mid x)} \right) > \mathrm{KL}(p^{(2)}(.|y_{<t}, x)||p^{(2)}(.|y_{<t}, x)) + \log \left( \frac{p(y_{<t} \mid x)}{p^{(2)}(y_{<t} \mid x)} \right) \tag{8}$$

$$= \log \left( \frac{p(y_{<t} \mid x)}{p^{(2)}(y_{<t} \mid x)} \right) > \log \left( \frac{p(y_{<t} \mid x)}{p^{(1)}(y_{<t} \mid x)} \right) = \mathrm{KL}(p^*||p^{(1)}(.|y_{<t}, x)) + \log \left( \frac{p(y_{<t} \mid x)}{p^{(2)}(y_{<t} \mid x)} \right). \tag{9}$$

Thus, $p^*$ cannot be the optimal solution, and thus $p^* = p^{(2)}(.|y_{<t}, x)$. Hence the second condition from Lemma 3.3 holds.

Finally, note that if $p^{(i)}(y_t|y_{<t}, x) = 0$ for some $y_t$, we necessarily have that $p^*(y_t) = 0$. In this case, the optimal solution may satisfy neither of the two conditions from Lemma 3.3.

# D    IMPLEMENTATION DETAILS

## D.1    COMPUTATIONAL RESOURCES

All experiments were conducted using NVIDIA A40 GPUs. For each token generated by CP-Fuse, the method involves two steps: (1) a forward pass through the two models, and (2) solving an optimization problem via grid search.

- *Forward Passes:* The base models perform a forward pass at each decoding step. In our experiments, both models were run on a single NVIDIA A40 GPU. With proper parallelization, the forward pass introduces no overhead compared to decoding with a single model.
- *Grid Search:* We evaluated the computational cost of the grid search in CP-Fuse. Using a grid size of 20 and varying the batch size, we measured the average overhead across 1,000 generated tokens, solving the optimization problem at each step. The results showed minimal overhead, as summarized in Table 15.

| Batch Size | Time (seconds) |
|---|---|
| 1 | $4.0 \times 10^{-4} \pm 4 \times 10^{-5}$ |
| 10 | $4.1 \times 10^{-4} \pm 3 \times 10^{-5}$ |
| 25 | $5.2 \times 10^{-4} \pm 3 \times 10^{-5}$ |
| 50 | $6.0 \times 10^{-4} \pm 3 \times 10^{-5}$ |

Table 15: Average grid search overhead for different batch sizes.

**Decoding Times**   In our experiments, the overall decoding speeds, averaged over 100 generations using LLaMA 2 7B models as described in the main paper, Appendix D.2 and Appendix D.3, are as follows:

- *Single model (1 GPU):* $16.25 \pm 0.64$ tokens/second

- *CP-Fuse (1 GPU):* $15.83 \pm 2.96$ tokens/second

**Training Times**   Fine-tuning the models for 50 epochs took approximately 7 to 9 hours each. We fine-tuned a total of 18 models for the experiments presented in the main paper and the Appendix.

## D.2   FINE-TUNING DETAILS

We fine-tuned our models using a setup inspired by the repository *finetuning-harness*, available under the MIT License[4]. The training was performed on A100 GPUs.

The main hyperparameters for our fine-tuning process are listed in Table 16. We fine-tuned our

Table 16: Main hyperparameters for fine-tuning

| Hyperparameter | Value |
|---|---|
| Sequence Length | 2048 |
| Batch Size | 1 |
| Learning Rate | 5e-5 |
| Gradient Accumulation Steps | 1 |
| Optimizer | AdamW (8-bit) |
| Warmup Steps | 50 |
| Neptune Noise | $\alpha = 5.0$ |

models with Neptune noise (Jain et al., 2023) set to $\alpha = 5.0$. We did not perform any low-rank adaptation.

For the overfitted models, we trained StarCoder for 50 epochs (both in experiments with the `Python instructions` and the `APPS` datasets), LLaMa2 for 50 epochs (both in `MathAbstracts` and `WritingPrompts`), Phi-2 for 50 epochs, and GPT-2 XL for 20 epochs.

## D.3   DECODING DETAILS

We decode with greedy search. For the code task, the maximum sequence length is 2048 tokens in the `Python instructions` dataset and 512 in the `APPS`, `MBPP`, and `HumanEval` datasets, and for the text task, it is 1024 tokens for all datasets. This configuration is used both for our method and CP-$\Delta$. For `APPS`, `MBPP`, and `HumanEval`, we base our implementation on the *bigcode-evaluation-harness* repository, available under the Apache-2.0 License[5].

---

[4]GitHub Repository
[5]GitHub Repository

### D.4 DATASETS

We use four code-based (`Python instructions`, `APPS`, `MBPP`, `HumanEval`) and two text-based (`MathAbstracts`, `WritingPromtps`) datasets in our experiments, all downloadable from *HuggingFace*.

The first code-based dataset[6] is an instructional dataset for Python (`Python instructions`), containing two types of tasks: (1) generating a description of a given code, and (2) generating code that solves a given task. For our experiments, we only consider instances of the latter. We removed the docstring from all instances since its content was repeated across samples, compromising our assumption on the separability of copyrighted content (Section 3.1). The `APPS` dataset[7] is a benchmark for code generation with 10,000 problems in Python. Each sample consists of a programming problem formulation in English, some ground truth Python solutions, and test cases. We sample random subsets for fine-tuning and evaluation for our experiments. Both `MBPP`[8] and the `HumanEval`[9] datasets are standard for assessing code generation, and follow a similar structure of natural language instructions and solutions in Python code. They contain 378 and 164 programming problems, respectively. We use the sanitized version of `MBPP`, `MBPP+`, and the instruction-based version of `HumanEval`, `InstructHumanEval`.

For the text-based experiments, we use the AutoMathText dataset[10] (Zhang et al., 2024b), referred to as `MathAbstracts`. This dataset compiles an extensive set of mathematical texts from arXiv, OpenWebMath, RedPajama, Algebraic Stack, and other sources, with titles generated by the state-of-the-art language model Qwen-72B[11]. Finally, the `WritingPrompts` dataset[12] (Fan et al., 2018) contains amateur-level stories from a Reddit forum. Prompts are story premises (titles) given to users, who then write the corresponding stories. We only keep one story per title in our dataset.

### D.5 BASELINES

**System Self-Reminder (SystemPrompt)**    For SystemPrompt, we simply preface the prompts with the text:

> "You are a helpful, respectful, and honest assistant. When generating your response, do not reproduce memorized content."

**MemFree**    We implement MemFree (Ippolito et al., 2023), setting $n = 10$ and using the fine-tuning sets as a blocklist.

**Goldfish Loss**    Finally, for the wrapping experiments (Section 4.3), we train the LLaMa2 models for 50 epochs using the Goldfish Loss (GL) method (Hans et al., 2024), with the dropout frequency set to $k = 16$. We implemented the hash-table-based strategy with a context width of 4.

### D.6 METRICS

#### D.6.1 COPYRIGHT INFRINGEMENT

**Exact Matching (EM)**    Exact Matching (EM) measures the length of the longest matching substring between the model's output and the ground truth text. This metric is useful for assessing how well the model captures continuous segments of the reference text.

**Infringement Count ($IC_k$)**    Infringement Count ($IC_k$) captures the number of k-grams (substrings of length k) in the model's output that have an exact match in the ground truth text. This metric

---

[6]Nan-Do/instructional_code-search-net-python

[7]`APPS` (Hendrycks et al., 2021)

[8]`MBPP` (Austin et al., 2021)

[9]`InstructHumanEval` (Chen et al., 2021)

[10]math-ai/AutoMathText

[11]Visit the GitHub repository for additional details.

[12]`WritingPrompts`

assesses the content similarity and potential for copyright infringement based on the number of matching k-grams.

**ROUGE-L (ROU)**   ROUGE (Recall-Oriented Understudy for Gisting Evaluation) measures the overlap of n-grams, word sequences, and word pairs between the generated text and reference text. It focuses on recall, assessing how much of the reference text is captured by the generated text. The value of ROUGE ranges from 0 to 1, where 1 indicates perfect recall. The most common variant, ROUGE-L, is computed based on the longest common subsequence (LCS):

$$\text{ROUGE-L} = \frac{\text{LCS}}{\text{length of reference text}}$$

We implement the ROUGE-L with the `rouge_score` package.

**BLEU (BLE)**   BLEU (Bilingual Evaluation Understudy) measures the overlap of n-grams between the generated text and reference text, with a penalty for shorter outputs. It is computed using a modified precision score that includes a brevity penalty to discourage overly short translations. The value of BLEU ranges from 0 to 1, where 1 indicates perfect precision. For this study, we use uniform weights for n-grams:

$$\text{BLEU} = \text{BP} \times \exp\left(\sum_{n=1}^{N} \frac{1}{N} \log p_n\right)$$

where BP is the brevity penalty, $p_n$ is the precision for n-grams, and $N$ is the highest order of n-grams considered. For our experiments, we average equally BLEU-1, BLEU-2, BLEU-3, and BLEU-4. We implement the BLEU with the `nltk` package with default hyperparameters and using the smoothing function.

**METEOR (MET)**   METEOR (Metric for Evaluation of Translation with Explicit ORdering) evaluates the alignment between the generated text and reference text by considering synonyms, stemming, and exact matches. Unlike BLEU, which focuses on n-gram precision and typically measures performance at the corpus level, METEOR emphasizes unigram recall and aims to better align with human judgment at the sentence level. The value of METEOR ranges from 0 to 1, where 1 indicates perfect alignment. It combines precision, recall, and a fragmentation penalty to account for the alignment of chunks. It is computed as:

$$\text{METEOR} = F_{mean} \times (1 - P)$$

where $F_{mean}$ is the harmonic mean of precision and recall, and $P$ is the fragmentation penalty. We implement the METEOR with the `nltk` package with default hyperparameters.

**Jaccard Similarity (JAC)**   Jaccard Similarity (JAC) measures the intersection over the union of the sets of words in the generated text and reference text. It provides a simple measure of similarity, indicating how many words are shared between the two texts relative to the total number of unique words. The value of Jaccard Similarity ranges from 0 to 1, where 1 indicates identical sets. It is computed as:

$$\text{Jaccard Similarity} = \frac{|A \cap B|}{|A \cup B|}$$

where $A$ and $B$ are the sets of words in the generated text and reference text, respectively.

**Cosine Similarity (COS)**   Cosine Similarity (COS) measures the cosine of the angle between the word vectors of the generated text and reference text. This metric assesses the similarity in the direction of the vectors, providing an indication of how similar the two texts are in terms of their overall content distribution. The value of Cosine Similarity ranges from 0 to 1, where 1 indicates perfect similarity.

**Semantic Similarity (SEM)**   Semantic Similarity (SEM) evaluates the similarity between the generated text and reference text using a semantic model. This metric captures the meaning and context of the texts, providing a measure of how well the model understands and replicates the underlying semantics of the reference text. The value of Semantic Similarity ranges from 0 to 1, where 1 indicates perfect semantic alignment. We compute the semantic similarity using the `spacy` package with the English pipeline optimized for CPU.

**Levenshtein Distance (LEV)**    Levenshtein Distance (LEV) measures the minimum number of single-character edits (insertions, deletions, or substitutions) required to change the generated text into the reference text. A higher Levenshtein distance indicates greater dissimilarity, while a lower distance indicates greater similarity. The value of the normalized Levenshtein Distance ranges from 0 to 1, where 0 indicates identical texts. We compute the normalized Levenshtein distance with a sliding window to handle cases where the lengths of the generated and reference texts differ significantly.

**JPlag (JP)**    JPlag (JP) (Prechelt et al., 2002) is a specialized software plagiarism detection tool that measures the similarity between two programs, producing a score ranging from 0.0 to 1.0. JPlag operates by converting each program into a stream of canonical tokens and then attempting to cover one token string with substrings taken from the other, using a method known as "Greedy String Tiling." This approach allows JPlag to go beyond simple byte-by-byte comparison, as it takes into account programming language syntax and program structure. As a result, it is robust against various obfuscation techniques that might be used to disguise similarities between plagiarized files. We use the implementation from the original repository[13].

**Dolos**    Dolos (Maertens et al., 2022) is a state-of-the-art plagiarism detection tool that scores the similarity between two programs on a scale from 0.0 to 1.0. The tool works by transforming the code into a more abstract representation, capturing essential structural elements while filtering out less significant details. This allows Dolos to detect plagiarism more accurately, even when the code has been obfuscated. The similarity score is computed using sophisticated algorithms that consider both the syntax and the semantics of the code. For a detailed description of the algorithm, we refer to the original paper (Maertens et al., 2022). We use the open-source implementation provided by the original developers[14].

### D.6.2    UTILITY

**Pass@1 (Pass at 1)**    Pass@1 evaluates the success rate of a model in generating a correct solution on its first attempt. Specifically, it measures the proportion of cases where the model's first output matches the correct solution. The value of Pass@1 ranges from 0 to 1, where 1 indicates that the model always generates a correct solution on the first attempt.

**Fluency**    We evaluate *fluency* using an LLM-as-judge framework (Zheng et al., 2024) based on the Prometheus-v2 model (Kim et al., 2024). Specifically, Prometheus-v2 serves as a judge for the stories generated by models fine-tuned on the `WritingPrompts` dataset. We adopt a five-point fluency rubric, similar to the one used in (Fu et al., 2023; Chen et al., 2024), as shown in Figure 21. The main difference is that we explicitly account for the amateur nature of the stories, acknowledging that they may contain some grammatical or spelling mistakes. Model-based fluency metrics have been shown to closely align with human evaluations (Liu et al., 2023; Sottana et al., 2023), especially for the `WritingPrompts` dataset (Chiang & Lee, 2023). Prometheus-v2 has also demonstrated consistent alignment with human annotators and GPT-4 (Kim et al., 2024).

**Perplexity (PPL)**    Perplexity (PPL) evaluates the quality of a language model. It measures how well a probability distribution or model predicts a sample. Lower perplexity indicates that the model is better at predicting the sample. The value of Perplexity ranges from 1 to infinity, where lower values indicate better performance.

---

[13]https://github.com/jplag/JPlag/
[14]https://github.com/dodona-edu/dolos

---

**Fluency Evaluation**

You are a fair judge assistant tasked with providing clear, objective feedback based on specific criteria, ensuring each assessment reflects the absolute standards set for performance.

**Task Description:** A story prompt, a story to evaluate, and a score rubric representing evaluation criteria are given.

1. Write detailed feedback that assesses the quality of the story strictly based on the given score rubric, taking into account that the story is from a non-professional Reddit forum (`WRITINGPROMPTS`).

2. After writing the feedback, assign a score that is an integer between 1 and 5, referring to the score rubric.

3. The output format should look as follows: `"(write feedback for criteria) [RESULT] (an integer number between 1 and 5)"`

4. Please do not generate any other opening, closing, or explanations.

**The prompt of the story:** `{title}`

**Story to evaluate:** `{story}`

**Score Rubrics:** Evaluate whether the given text is well-written and grammatical, considering the casual, non-professional nature of the forum.

- **Score 1:** The response is very unclear, with significant grammatical errors that make comprehension difficult.

- **Score 2:** The response has noticeable grammatical issues or disorganization, which may affect reader comprehension but is still somewhat understandable.

- **Score 3:** The response is generally understandable, with some minor errors or awkward phrasing that could be improved.

- **Score 4:** The response is clear and easy to follow, with only minor grammatical issues expected in a casual setting.

- **Score 5:** The response is very clear and well-organized, considering the informal nature of the forum.

**Feedback:** `[Your feedback here]`

Figure 21: Fluency rubric for Prometheus-v2

