# OpenReview forum: "Copyright-Protected Language Generation via Adaptive Model Fusion"
_ICLR.cc/2025/Conference — ICLR 2025 Oral_

### Official Review · Reviewer_BxKD · 2024-10-29

**Soundness:** 3
**Presentation:** 4
**Contribution:** 3
**Rating:** 8
**Confidence:** 3

**Summary:**

This paper presents the CP-Fuse algorithm, a decoding-time strategy to reduce the amount of verbatim memorization (of copyright data) during LLM decoding. The algorithm operates in two simple steps:

1. Two models are trained on potentially overlapping subsets of the training data, as long as the copyright part is well separated between the two subsets.

2. The two models are interpolated during inference, with a dynamic per-token weighting scheme that ensures an equal contribution from each model towards the output.

The authors conduct experiments in a number of domains (writing, coding, math abstracts) and find that models trained with CP-Fuse are significantly less prone to verbatim memorization. Moreover, the fused models continue to preserve the quality of the overfit models which did not undergo any interpolation.

**Strengths:**

1. The paper presents good contributions in an important research area: preventing verbatim reproduction of copyright information. This topic is getting extremely important recently due to the presence of copyright data in frontier models, which has led to many multi-million dollar lawsuits.

2. The proposed algorithm is intuitive and effective --- it reduces reproduction of copyright material by 25x, and outperforms competing algorithms like CP-delta and MemFree. Experiments are conducted in four different domains using a variety of pretrained models (LLAMA2, GPT-2 XL, Phi)

3. Beyond the main experiments, the paper has a lot of good analysis including (1) a combination of CP-Fuse with training time copyright protection methods; (2) robustness to long prefix probing attacks; (3) different decoding strategies; (4) interpolation of three models; (5) analysis of the chosen interpolation weights.

4. The paper has a great appendix with a lot details and model outputs from both the overfitted model and CP-Fuse. Overall, the paper was well presented, and the figures were clear and easy to understand.

**Weaknesses:**

I had no concerns regarding the copyright prevention aspect of the algorithm, and thought the experiments were well designed. However, almost all my concerns were regarding the potential quality / utility drop due to interpolation of LLM probabilities during inference. Please let me know if some subset of the weaknesses / questions was covered somewhere in the appendix.

1. **Measuring the quality loss due to CP-Fuse on stronger baselines** would make the paper stronger. In Table 2, currently all models have a HumanEval pass@1 score of less than 29%, which is far lower than the state-of-the-art (Claude 3.5 Sonnet reached 93.7% recently). My hypothesis is that the quality drop due to LLM interpolation maybe higher / more visible with stronger LLMs. Could this method be applied to the larger LLAMA-3 or Gemma-2 family of models?

2. **Unclear quality tradeoff between multitask learning and fusing during decoding**. A lot of the model merging suggests a quality drop from multitask learning to merging models trained on individual datasets. What is the performance difference between a single model trained on `D1+D2` vs CP-Fuse of a model trained on `D1` with a model trained on `D2`?

3. **Experimental settings of overfitting on a small dataset a bit artificial**. The current experimental setting of overfitting on a single small domain-specific dataset seems a bit artifical. In practice, SFT models are trained on a large mixture of multiple tasks (like TULU-v2 https://arxiv.org/abs/2311.10702) and not overfit on any particular task. I would be curious to better understand: (i) what is the recommended strategy for dataset splitting when many source datasets are involved? (ii) Same question as weakness 2, what is the quality tradeoff between multi-tasking and running CP-Fuse on two models?

4. **Is there a slowdown in decoding time?** The algorithm involves sampling from two LLMs at inference, and estimating interpolation weights. How much does this slow down decoding time vs a vanilla single LLM sampling? What is the relative time taken between sampling and interpolation weight grid search?

**Questions:**

1. It wasn't super clear to me if the CP-Fuse theoretically guarantees copyright prevention to some n-gram threshold (example: 5-gram sequences will NEVER be copied irrespective of the learnt distributions). Looking at some examples (Figure 13) and Figure 3, I'm guessing this is not true --- But I'm wondering if the authors did any analysis on this / have any insights.

2. How does the method interact with RLHF? Would you suggest splitting RL prompts in a similar manner as done on SFT data? RLHF model outputs are known to be more deterministic than SFT model outputs (https://arxiv.org/pdf/2310.06452) --- does this mean the fusing algorithm will be less effective on RLHF'ed models?

3. Nit: L107: does not assign non-negligible probability --> assigns negligible probability

---

> ### Author Response · Authors · 2024-11-19
> **Rebuttal**
>
> We thank the reviewer for recognizing the relevance of the problem we tackle with CP-Fuse and appreciate their positive feedback on our additional analysis and appendices. In our general comments, we have addressed concerns regarding potential slowdown in decoding time.
>
> **CP-Fuse for Stronger Baselines, e.g., LLama 3**
>
> We replicated the code generation experiments from the main paper (originally conducted with the StarCoder model) using LLaMA 3 to assess CP-Fuse’s performance with a stronger baseline and a precise utility measure via the pass@1 score. Specifically, we fine-tuned two LLaMA 3 base models using LoRA on separate splits from the APPS dataset, training each for 5 epochs (as opposed to 50 epochs in the main paper). We then evaluated first-shot test passing rates on HumanEval, APPS, and MBPP, comparing CP-Fuse to a single LLaMA 3 model trained on the combined splits without CP-Fuse.
>
> | Model | HumanEval | APPS | MBPP
> |-----|-----|-----|-----|
> | Single Model | 0.34 | 0.46 | 0.54
> | CP-Fuse | 0.40 | 0.52 | 0.52
>
> As in the main paper, CP-Fuse shows no utility drop compared to the single model.
>
> **Comparison of CP-Fuse with a Single Model**
>
> The results above, along with the following from the StarCoder (code generation) and LLaMA 2 (fluency evaluation on WritingPrompts), confirm that CP-Fuse maintains comparable utility to a model trained on the full dataset.
>
> | Model | HumanEval | APPS | MBPP | Fluency (WritingPrompts) |
> |-----|-----|-----|-----|-----|
> | Single Model | 0.29 | 0.47 | 0.44 | 2.17 |
> | CP-Fuse | 0.28 | 0.47 | 0.43 | 2.17 |
>
> **Recommended Strategy for Splitting Datasets and CP-Fuse on Multiple Tasks**
>
> Our recommended strategy is to duplicate datasets for tasks that are not copyright-sensitive and use them to train all models, while partitioning sensitive tasks so that sensitive content only appears in one model's training data. This ensures that each model can independently perform well on the tasks, so merging them with CP-Fuse does not result in the performance drop that the reviewer was concerned about.
>
> Following this approach, we conducted additional experiments with LLaMA 3, fine-tuning two base models on the APPS dataset and then on disjoint splits of (1) MathAbstract, (2) WritingPrompts, and (3) Alpaca general knowledge [1]. In these tests, we observed no quality drop in pass@1 scores on HumanEval, APPS, and MBPP after fine-tuning the additional datasets. Importantly, CP-Fuse combining the two models achieves scores similar to those of the single models.
>
> **Questions**
>
> *Is there an n-gram threshold?* There is no specific n-gram threshold. If a sequence is highly probable under both models, CP-Fuse can reproduce it by assigning similar weights to each model's distribution during fusion. Conversely, if only one model has seen the sequence, the reproduced length depends on how "common" it appears under the second model, even if it wasn’t part of its training data. The evolution of these weighting parameters across example sequences is shown in Figure 7
>
> *Will CP-Fuse be less effective on RLHF?* The interaction between memorization and RLHF remains underexplored. The only study we are aware of is [2], which examines this issue in code generation and concludes that RLHF significantly reduces the likelihood of memorizing data used for reward modeling and reinforcement learning compared to SFT. The authors of [2] further state, "Because of this low risk, it may be possible to leverage sensitive and/or proprietary data during reward modeling." Therefore, we expect copyright infringement to be less of a concern in this setup. However, CP-Fuse could still be applied, and we anticipate it would exhibit similar behavior as with SFT.
>
> [1] Taori, R., Gulrajani, I., Zhang, T., Dubois, Y., Li, X., Guestrin, C., ... & Hashimoto, T. B. (2023). Stanford alpaca: an instruction-following llama model (2023). URL https://github. com/tatsu-lab/stanford_alpaca, 1(9)
>
> [2] Pappu, A., Porter, B., Shumailov, I., & Hayes, J. (2024). Measuring memorization in RLHF for code completion. arXiv preprint arXiv:2406.11715.

---

> > ### Comment · Reviewer_BxKD · 2024-11-21
> > **Thank you for the response! Updated score to 8**
> >
> > Thank you for the detailed response and addressing many of my concerns! I've updated the score to 8

---

### Official Review · Reviewer_87cP · 2024-10-31

**Soundness:** 3
**Presentation:** 4
**Contribution:** 3
**Rating:** 8
**Confidence:** 3

**Summary:**

The paper introduces Copyright-Protecting Model Fusion (CP-Fuse), a technique designed to reduce the likelihood of language models reproducing copyrighted material during text generation. CP-Fuse combines outputs from different language models, each trained on distinct copyrighted datasets, using an adaptive fusion mechanism. This method mitigates the risk of copyright infringement by limiting memorization without compromising content quality.

**Strengths:**

This is a very well-written paper that is easy to follow. It introduces a new, adaptable method for copyright protection that can be combined with other training techniques to further increase its efficacy. A big strength is that the authors provide an extensive evaluation across multiple methods to measure memorization and they also evaluate the fluency/quality of the model outputs.

**Weaknesses:**

The approach assumes that copyrighted material can be effectively separated into distinct datasets, a process that becomes challenging at larger scales and real-world scenarios. This limitation is noted in the Appendix, but the authors only address the practical challenges of creating copyrighted datasets, and not the potential ethical issues and dual-use concerns that may arise from using such data and models.

Moreover, even though the authors mention how previous works are computationally heavy, they do not include a statement detailing the computational resources used in their experiments.

**Questions:**

I was not sure if this should be added as a weakness or question, but have you considered that your approach might not be effective for adversarial prompts? How does CP-Fuse handle cases of minor text modifications where the original structure is still recognizable? Did you try any adversarial extraction methods (similar to the goldfish paper)?

I recommend specifying the duration of each experiment, the hardware (e.g., type of GPU) employed, and the overall computation time to provide clarity on the method's computational feasibility and sustainability.

It is also necessary to address any anticipated risks or ethical concerns associated with this work. The prerequisite of having a dataset with copyright material and a model trained on such data needs to be further discussed.

---

> ### Author Response · Authors · 2024-11-19
> **Rebuttal**
>
> We thank the reviewer for their positive comments on our experimental evaluation and the clarity of our presentation. In our general comments, we have addressed the issues regarding the separability assumption and computational complexity.
>
> **Potential Ethical and Dual-Use Concerns**
>
> Thank you for highlighting this important point. We acknowledge that real-world applications of CP-Fuse could face legal restrictions related to dataset and model usage. Copyright protection for LLMs is complex and lacks clear legislative consensus on what constitutes infringement. We agree on the importance of discussing these concerns and will add a paragraph in the updated draft to acknowledge them, while refraining from specific recommendations, as these are the responsibility of organizations' legal departments.
>
> **Questions**
>
> *Adversarial Attacks*: The reviewer makes a valid point. While the balancing property in CP-Fuse provides some intuitive protection, our method could still be susceptible to adversarially crafted prompts. We envision CP-Fuse being used primarily in controlled environments, such as API setups, where only black-box attacks are feasible. We attempted GCG-like attacks [1] in this setting, but they were too computationally demanding and did not yield successful results given our resources. We believe future work could explore stronger attacks.
>
> *Experimental Details*: Thank you for highlighting this. We have reported our infrastructure details (NVIDIA A40 GPUs) and CP-Fuse runtimes in the General Comments. We will include these, along with additional details on fine-tuning and other experimental settings, in the updated draft.
>
> *Ethical Concerns with Datasets and Models*: As noted above, we will address ethical considerations related to model and data use. Additionally, we clarify that all datasets and models used in our experiments are publicly available from Hugging Face and do not involve real-world copyright-protected material.
>
> [1] Zou, A., Wang, Z., Carlini, N., Nasr, M., Kolter, J. Z., & Fredrikson, M. (2023). Universal and transferable adversarial attacks on aligned language models. arXiv preprint arXiv:2307.15043.

---

> > ### Author Response · Authors · 2024-11-24
> > **Follow-up**
> >
> > Dear Reviewer 87cP,
> >
> > We hope our rebuttal has addressed your concerns and answered your questions. As the end of the discussion period approaches, we would like to ask if you have any additional questions or concerns, particularly regarding our rebuttal.
> >
> > Thank you once again for your time and thoughtful feedback!

---

> > > ### Comment · Reviewer_87cP · 2024-11-24
> > > **Official Comment by Reviewer  87cP**
> > >
> > > Thank you for the clarifications. I acknowledge I have read your response.

---

### Official Review · Reviewer_MfZh · 2024-11-04

**Soundness:** 4
**Presentation:** 4
**Contribution:** 3
**Rating:** 8
**Confidence:** 3

**Summary:**

This paper addresses a significant concern in the field of large language models (LLMs) — the risk of generating copyrighted material. The authors propose a novel inference-time method called Copyright-Protecting Model Fusion (CP-Fuse), which reduces the likelihood of reproducing copyrighted content without impacting the quality of the generated text. CP-Fuse achieves this by adaptively combining the outputs from multiple models, each trained on disjoint datasets containing distinct copyrighted material. By leveraging the balancing property in fusion, the paper claims to prevent any single model from dominating the output, thereby mitigating the regurgitation of memorized, copyright-protected text. The experimental results demonstrate CP-Fuse’s effectiveness compared to existing inference-time copyright-protection techniques, with improvements in copyright infringement reduction while maintaining the utility of generated outputs.

**Strengths:**

The paper is well-written and highlights the critical issue of copyright protection in LLMs, which is increasingly relevant as these models are deployed in diverse domains. Addressing this issue is essential for responsibly advancing generative AI.

CP-Fuse introduces a fresh approach to copyright protection by employing model fusion, which differentiates it from conventional methods focused solely on filtering or training-time constraints. The adaptive fusion strategy shows promise in providing a feasible balance between copyright compliance and output quality.

**Weaknesses:**

While the fusion of multiple models appears to improve copyright protection, it may introduce potential inefficiencies, especially at inference time. However, the paper lacks experimental results that measure the efficiency or computational cost of CP-Fuse compared to single-model approaches. Such evaluations are important to gauge the method's practicality in real-world applications.

The paper evaluates utility on task-specific datasets but does not test CP-Fuse on general benchmarks like MMLU. Adding such benchmarks could provide a more comprehensive view of the trade-offs in generalization and utility across diverse language tasks.

As an inference-time approach, CP-Fuse cannot prevent malicious users from extracting model parameters to obtain copyrighted content. If a model has already been trained on multiple datasets containing copyrighted material, an attacker could use a single dataset-focused fine-tuned model to retrieve specific copyrighted documents easily. This limitation suggests that CP-Fuse may not provide sufficient protection against all types of copyright violations, particularly those involving parameter access or model reuse by adversaries.

**Questions:**

Is model fusion essential to address copyright concerns, or could other strategies, potentially simpler, also achieve comparable results? It would be helpful to know if the authors considered or tested any alternative methods.

The paper’s extensive use of various metrics is commendable; however, previous works have often relied on the Longest Common Subsequence (LCS) to gauge memorization. Was there a specific reason LCS was excluded from the analysis?

The balancing property is central to the proposed fusion strategy, but is its relevance to copyright protection empirically validated? While balancing prevents any single model from fully influencing the generation, it is not immediately clear if or how this directly contributes to reducing copyright infringement. Further clarification on its role and effectiveness would be beneficial.

How does CP-Fuse relate to Mixture of Experts (MoE) models?

---

> ### Author Response · Authors · 2024-11-19
> **Rebuttal**
>
> We thank the reviewer for recognizing the relevance and promise of our method and for their positive feedback on the clarity and quality of the manuscript. In our general comments, we have addressed concerns about potential inefficiencies of CP-Fuse.
>
> **Testing CP-Fuse on MMLU**
>
> Thank you for this suggestion. Initially, we did not conduct experiments on multi-class classification because copyright infringement in such a setting is less clearly defined. Instead, we focused on text and code generation. For completeness, we have now conducted a 5-shot MMLU evaluation using models fine-tuned on the MathAbstracts dataset, as described in the paper. The single models achieved a score of 44.7 (consistent with values reported in the literature [1]), while CP-Fuse combining the two models yielded a score of 44.9. These results indicate that CP-Fuse does not degrade utility in this classification setting.
>
> **CP-Fuse Under Attacks Involving Parameter Access**
>
> We acknowledge the reviewer's observation that CP-Fuse, as an inference-time strategy, relies on underlying models that could risk copyright infringement without protection. We envision CP-Fuse's primary application in API-like interactions or scenarios where users do not have access to model parameters. We will clarify this threat model in the updated draft. Additionally, as demonstrated in Section 4.3, CP-Fuse can be combined with other methods to guard against more powerful, white-box attacks.
>
> **Questions**
>
> *Alternative Protection Strategies*: We consider model fusion a promising approach to copyright protection and are unaware of alternative methods with similar properties to CP-Fuse (e.g., the balancing property) that operate with a single model. Our literature review primarily identified filtering strategies, which have significant limitations, as discussed in Section 4.2.
>
> *Why LCS Was Excluded from Analysis*:  If the reviewer refers to the longest contiguous subsequence between two sequences, this is what we denote as the exact matching (EM) metric (see Appendix D.5.1 for a description of all metrics). Could the reviewer confirm if this is the metric in question?
>
> *Relevance of Balancing Property*: We approach copyright protection from a verbatim (and quasi-verbatim) reproduction perspective. In Section 3.3 and Appendix A.7, we detail how the balancing property, under our separability assumptions, helps prevent such infringement. Our experiments further validate its effectiveness.
>
> *Connection to MoE*: There is indeed a connection between CP-Fuse and Mixture of Experts, particularly in how both dynamically select which model to "activate"—MoE based on the input prompt and CP-Fuse based on the history of the currently decoded sequence.  However, MoE focuses on efficiency and model capacity, whereas CP-Fuse targets copyright protection.
>
> [1] Touvron, H., Martin, L., Stone, K., Albert, P., Almahairi, A., Babaei, Y., ... & Scialom, T. (2023). Llama 2: Open foundation and fine-tuned chat models. arXiv preprint arXiv:2307.09288.

---

> > ### Comment · Reviewer_MfZh · 2024-11-19
> >
> > Thank you and after reviewing the response, I have updated my assessment.

---

### Official Review · Reviewer_QpAc · 2024-11-04

**Soundness:** 3
**Presentation:** 3
**Contribution:** 2
**Rating:** 6
**Confidence:** 3

**Summary:**

This paper introduces the Copyright-Protecting Model Fusion (CP-Fuse) as a mitigation approach for copyright infringement in inference-time language generation. CP-Fuse adaptively aggregates the outputs of models trained on disjoint sets of copyrighted material with the goal of minimizing the reproduction of copyrighted content. It also optimizes a balancing property that ensures no single model dominates. Extensive experiments on text and code generation tasks demonstrate that CP-Fuse significantly reduces copyright infringement while maintaining high output quality. The method is also shown to be robust against extraction attacks and compatible with other copyright-protection techniques.

**Strengths:**

## **1. The approach is novel and sound**

CP-Fuse introduces the more explored inference-time model fusion techniques to the recently emergent challenges of copyright infringement prevention, the application is novel and the problem addressed is important. The efficacy of the method is justified with both theoretical and empirical backing.

## **2. Extensive experiments on multiple use cases across scenarios**

The authors conduct thorough experiments across different datasets and scenarios (text, code, overfitted models). CP-Fuse consistently outperforms baselines like MemFree and CP-$\Delta$ in copyright protection metrics and achieves near "non-plagiarism" on the code task. Further evaluation results also show that CP-Fuse is robust against data extraction techniques, and can be integrated with other training-time copyright mitigation methods.

## **3. Output utility is high**

The paper reports that CP-Fuse does not sacrifice output quality, achieving comparable scores in fluency (for text) and accuracy (for code generation) relative to overfitted models.

**Weaknesses:**

## **1. The strong assumption of copyright separability**

The approach relies on the assumption that copyrighted content is separable across datasets and the user of CP-Fuse has access to multiple models trained on these separate datasets. This assumption may not hold for all real-world datasets, potentially limiting the method’s applicability.

## **2. The computational complexity might limit scalability**

The computational complexity of fusion with grid search parameters may present challenges, especially for larger models or longer text generations. Also, it remains unclear to me whether and how well this approach can scale over two disjoint datasets and models, given the increased computational complexity.

## **3. Limited scope of evaluation**

Currently, CP-Fuse is tested on relatively controlled datasets and tasks, the effectiveness in real-world applications with larger, diverse corpora containing complex copyright dependencies is not fully explored.

**Questions:**

1) Can you explain more about the impact of the separability assumption in real-world applications? How would CP-Fuse perform in scenarios where copyrighted materials cannot be easily isolated?

2) How does the performance of CP-Fuse scale with larger language models and increased amount of copyright materials? Would additional computational resources be required, or could there be potential optimizations?

3) For future works, could CP-Fuse be generalized to other modalities beyond text and code (e.g., images or audio)? If so, what adaptations might be necessary?

---

> ### Author Response · Authors · 2024-11-19
> **Rebuttal**
>
> We thank the reviewer for acknowledging the novelty of our method and the thoroughness of our experimental evaluation. In our general comment, we have addressed concerns regarding the separability of the copyright assumption and computational complexity.
>
> **Effectiveness in Real-World Scenarios**
>
> We recognize that the effectiveness of CP-Fuse in real-world applications requires further exploration. Our experiments focus on verbatim (and quasi-verbatim) reproduction of protected material as a straightforward case of potential copyright infringement. However, copyright protection for LLMs currently lacks clear legislative definitions—ongoing lawsuits have yet to reach conclusions [1, 2]. Establishing concrete guidelines for real-world applications is beyond the scope of our work. If the reviewer has specific use cases, tasks, or datasets in mind, we welcome suggestions for additional experiments.
>
> **Questions**
>
> *Impact of Separability Assumption*: Addressed in the General Comments.
>
> *Scaling CP-Fuse to Larger or More Models*: We have partially addressed this in the General Comments.  While we currently lack resources to scale CP-Fuse to very large models (e.g., 70B parameters), efficient parallelization can mitigate inefficiencies. The main limitations are communication overhead between GPUs and solving the grid search, which remains extremely fast (~$10^{-4}$ seconds for large batch sizes). Adding more models is up to practitioners; two models already offer significant protection, and adding more may require extra hardware and increase communication overhead.
>
> *Scaling CP-Fuse to More Copyrighted Data*: Our experiments already simulate an extreme case by treating all data as copyright-protected, requiring the model to avoid reproducing any training samples even after overfitting.
>
> *Generalization to Other Modalities*: This is an exciting question! CP-Fuse can be adapted to other autoregressive generative models, such as those used in image generation. It extends to models that use decoding algorithms for selection from a discrete set (pixels, audio frames, etc.) without significant modifications. Our experiments demonstrate merits in text and code; we leave exploration in other modalities for future work.
>
> [1]  Grynbaum, M. M., & Mac, R. (2023). The Times sues OpenAI and Microsoft over AI use of copyrighted work. The New York Times, 27.
>
> [2] Brittain, B. 2023. Pulitzer-winning authors join OpenAI, Microsoft copyright lawsuit. Reuters.

---

> > ### Author Response · Authors · 2024-11-24
> > **Follow-up**
> >
> > Dear Reviewer QpAc,
> >
> > We hope our rebuttal has addressed your concerns and answered your questions. As the end of the discussion period approaches, we would like to ask if you have any additional questions or concerns, particularly regarding our rebuttal.
> >
> > Thank you once again for your time and thoughtful feedback!

---

> > > ### Author Response · Authors · 2024-11-29
> > >
> > > Dear Reviewer QpAc,
> > >
> > > We appreciate the opportunity provided by the extended deadline to follow up. If there are any additional clarifications or points you would like us to address, we would be happy to do so before the discussion period concludes.
> > >
> > > Best regards,
> > > The authors

---

### Author Response · Authors · 2024-11-19
**General Comment**

We express our sincere thanks to the reviewers for their detailed review. It is encouraging to see that
1. **all reviewers agreed on the relevance of the problem we address**, namely copyright protection for language models, and recognized the **extensive experimental evaluation** of multiple datasets and tasks.
2. Furthermore, we are pleased that they found our approach **“a fresh, different take from other methods**, showing promise in addressing copyright while maintaining utility,” (Rev MfZh) and the proposed algorithm to be **“intuitive and effective”** (Rev BxKD).
3. It is great to hear that they found the manuscript **“very well-written and easy to follow”** (Rev 87cP) and noted that it **“has a lot of good analysis beyond the main experiments”** (Rev BxKD).

We also appreciate the constructive feedback aimed at refining our manuscript. The following comments address prominent issues highlighted by the reviewers. We will attend to specific feedback in our individual responses.

**Increased computational complexity**

We appreciate the reviewers' concerns regarding the computational complexity of CP-Fuse. To clarify, CP-Fuse involves two steps for generating each token: (1) a forward pass for the two models and (2) solving an optimization problem via grid search.

*Forward Passes:* These can be straightforwardly parallelized. In our experiments, we fit the two models on a single NVIDIA A40 GPU. With correct parallelization, the forward pass introduces no additional overhead compared to decoding with a single model.

*Grid Search:* We evaluated the computational cost of the grid search in CP-Fuse. Using a grid size of 20—sufficient for our experiments—and varying the batch size, we measured the average overhead across 1,000 generated tokens, solving the optimization problem at each step. The results show minimal overhead:

| Batch size | Time (s) |
| -------- | ------- |
| 1 | $4 \times 10^{-4} ± 4 \times 10^{-5}$ |
| 10 | $4.1 \times 10^{-4} ± 3 \times 10^{-5}$ |
| 25 | $5 \times 10^{-4} ± 3 \times 10^{-5}$ |
| 50 | $6 \times 10^{-4} ± 3 \times 10^{-5}$ |

In our experiments, the overall decoding times averaged over 100 generations, using LLaMA 2 7B models (without low-rank adaptations), are as follows:

* **Single model (1 GPU)**: 16.25 ± 0.64 tokens/second
* **CP-Fuse (1 GPU)**: 15.83 ± 2.96 tokens/second

When running CP-Fuse on a single GPU, latency remains close to that of a single model. However, the increased memory requirements of CP-Fuse may necessitate additional hardware. In multi-GPU setups, latency could rise due to communication overhead—a common trade-off in ensemble techniques. Specific inefficiencies will depend on the models and hardware used, not CP-Fuse. We believe that the minimal increase in latency is a manageable trade-off given the copyright safeguards CP-Fuse provides.

**Separability of copyright assumption**

To address this concern, we conducted additional experiments introducing controlled overlap between two splits of the WritingPrompts dataset.

*Experimental Setup*: We split the dataset into two subsets of 1,000 examples each, introducing overlaps of 0%, 10%, 33%, and 66%. We considered a ‘’worst-case’’ scenario where overlapping data are exact duplicates—though in real-world settings, duplicates are unlikely to be exact. We trained two separate LLaMA 2 models on these subsets for 50 epochs, as in the main paper. For each level of overlap, we computed the copyright-protection metrics for split 1 (similar results were observed for split 2). The experiment was repeated with three random seeds, and we reported the worst-case results—instances with the highest potential for copyright infringement.

|Overlap | EM | ROU | BLE | LEV |
| -------- | ------- | -------- | ------- | -------- |
| 0.0 | 25.50 | 0.19 | 0.03 | 0.70 |
| 0.1 | 30.2 | 0.19 | 0.04 | 0.70 |
| 0.33 | 47.70 | 0.22 | 0.06 | 0.72 |
| 0.66 | 747.6 | 0.87 | 0.84 | 0.12 |

Even with a 10% overlap, metrics show only a slight increase compared to no overlap, and at 33%, protection remains reasonable on average. However, we acknowledge the lack of guarantees, as top copied sequences might include long verbatim segments. Analyzing worst-case scenarios—such as the longest potential copied segments as overlap increases—could improve applicability to safety-critical cases and is left for future work.

We recognize this assumption as a limitation and a key challenge in copyright protection. Alternative approaches, such as filtering, rely on blocklists, while training methods treating all data as protected often reduce utility, as seen with Differentially Private training. CP-Fuse is the first inference method to theoretically explain how reproducing protected material is unlikely, leveraging its balancing property. We encourage further theoretical and methodological work to address these challenges.

---

> ### Author Response · Authors · 2024-11-20
> **Updated Draft**
>
> We have updated the draft with the following changes:
>
> 1. We added Section B in the Appendix, "Discussion on the Separability of Copyright Assumption" (previously "Limitations"), where we further discuss the assumption and include the points discussed in the reviews: experiments with partial overlap, ethical limitations, and recommended strategies for splitting datasets and tasks.
>
> 2. We added Section D.1, "Computational resources," with a discussion on the additional overhead due to the grid search and details on the infrastructure, fine-tuning, and decoding times.
>
> 3. In Section 4.4, we specified the threat model and the nature of the adversary one would expect in such a setup.
>
> Please let us know if we have overlooked anything.

---

### Meta-Review · Area_Chair_WPTT · 2024-12-19

**Metareview:**

**Summary**

This paper focuses on a specific problem of LLMs. As these models memorize, they tend to emit what they have inside. Hence, the paper introduces a novel model to mitigate copyright infringement in inference-time language generation: Copyright-Protecting Model Fusion (CP-Fuse)

**Strengths**

- novelty of the approach
- extensive experimentation

**Weaknesses**

- Complexity and scalability of the approach
- As an inference-time approach, CP-Fuse cannot prevent malicious users from extracting model parameters to obtain copyrighted content.

**Final remarks**

The paper is clearly novel according to the reviewers and focus on an important issue of LLMs.

**Additional Comments On Reviewer Discussion:**

The authors did not answer to a specific comment of reviewer *MfZh* :
"As an inference-time approach, CP-Fuse cannot prevent malicious users from extracting model parameters to obtain copyrighted content."
This comment should be addressed in the final revision as it will hinder the approach that has been proposed.

---

### Decision · Program_Chairs · 2025-01-22

Accept (Oral)